# Simultaneous epigenomic profiling and regulatory activity measurement using e2MPRA

Zicong Zhang [1], Ilias Georgakopoulos-Soares [2], Guillaume Bourque [1,3,4,5], Nadav Ahituv [6,7] ✉ & Fumitaka Inoue [1] ✉

Using various biochemical assays that identify transcription factor (TF) binding and histone modifications, *cis*-regulatory elements (CREs) can be annotated in a genome-wide manner. However, these assays are descriptive and require functional validation. To the best of our knowledge, no technology can simultaneously analyze the regulatory function and epigenomic modifications of a specific sequence. Here, we develop an enrichment followed by epigenomic profiling massively parallel reporter assay (e2MPRA). This technique uses lentivirus to enrich for the integration of specific CREs into the genome and applies MPRA, Cut&Tag or ATAC-seq on them enabling simultaneous, high-throughput analysis of regulatory activity, protein binding, and epigenetic modification. We demonstrate that e2MPRA can dissect the epigenetic functions of TF motifs arranged within synthetic enhancers and evaluate the effects of sequence perturbation on epigenetic states. In summary, e2MPRA advances our understanding of the regulatory code, its effect on the epigenome and how its alteration leads to phenotypic effects.

Gene expression is regulated in a spatiotemporal manner by active *cis*-regulatory elements (CREs), such as promoters, enhancers, insulators and silencers. Nucleotide variants in CREs can alter gene expression, which can lead to subsequent changes in phenotypes, including cell differentiation, disease, evolution, and other biological phenomena. Active regulatory elements such as promoters and enhancers are bound by various transcription factors (TFs) that play different roles in transcriptional regulation, such as pioneering activity that induces chromatin accessibility, transcriptional activation and repression, histone modification, and chromatin remodeling activities that mediate higher-order chromatin structure.

Several technologies have been developed to identify CREs in a genome-wide manner. For example, DNase-seq and ATAC-seq can identify open chromatin regions that are accessible to TFs[1,2]. ChIP-seq identifies the binding of specific TFs, co-factors or histone marks by using chromatin immunoprecipitation followed by sequencing[3]. Cleavage Under Targets and Release Using Nuclease (CUT&RUN) uses an antibody to target specific TFs, co-factors or histone marks followed by the binding of a Protein A/G fused to micrococcal nuclease (pAG-MNase) to cleave the primary antibody-bound sites allowing to profile candidate CREs (cCREs) using a lower cell number than ChIP-seq[4,5] (i.e. at least 100 cells for a histone modification and 1,000 cells for a transcription factor are required). CUT&Tag (Cleavage Under Targets and Tagmentation) has been developed from CUT&RUN by using Tn5-mediated tagmentation, generating fragments ready for PCR enrichment and DNA sequencing[6]. These

[1]Institute for the Advanced Study of Human Biology (WPI-ASHBi), Kyoto University, Kyoto, Japan. [2]Institute for Personalized Medicine, Department of Biochemistry and Molecular Biology, The Pennsylvania State University College of Medicine, Hershey, PA, USA. [3]Department of Human Genetics, McGill University, Montréal, QC, Canada. [4]Victor Phillip Dahdaleh Institute of Genomic Medicine at McGill University, Montréal, QC, Canada. [5]Canadian Center for Computational Genomics, McGill University, Montréal, QC, Canada. [6]Department of Bioengineering and Therapeutic Sciences, University of California San Francisco, San Francisco, CA, USA. [7]Institute for Human Genetics, University of California San Francisco, San Francisco, CA, USA. ✉e-mail: nadav.ahituv@ucsf.edu; inoue.fumitaka.7a@kyoto-u.ac.jp

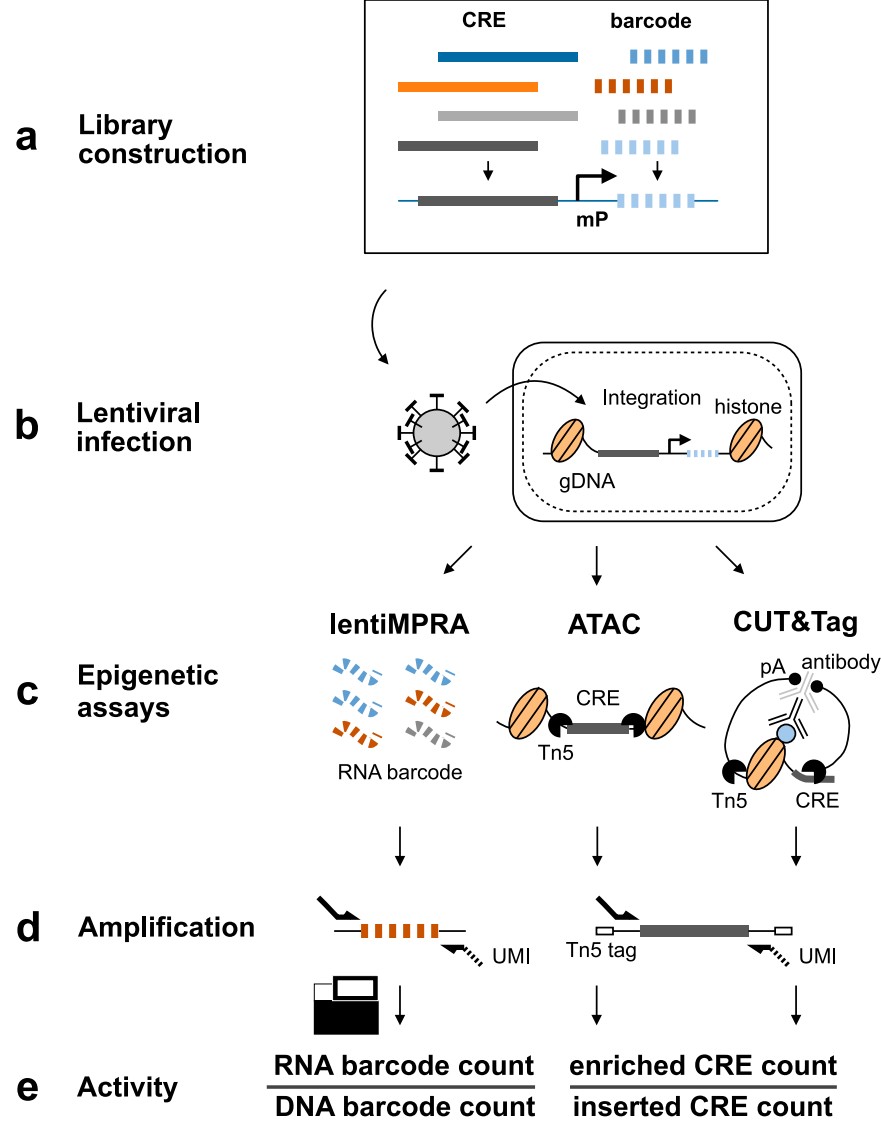

**Fig. 1 | e2MPRA overview. a** Designed cCRE libraries are associated with barcode sequences and cloned into reporter plasmids. **b** The plasmid libraries are packaged into lentivirus and infected into cells to allow genomic integration and enrichment for epigenetic analyses. **c** RNA barcodes (for lentiMPRA) and Tn5-tagmented DNA fragments (for ATAC and CUT&Tag) are extracted from infected cells. Genomic DNA is also extracted to estimate integration frequency. **d** Extracted samples are amplified with unique molecular identifiers (UMIs) and sequenced to quantify genomic integration frequency for each cCRE and regulatory and epigenetic activity. **e** LentiMPRA activity is calculated by RNA/DNA barcode count. For ATAC and CUT&Tag, epigenetic activity is calculated by dividing the number of Tn5-tagged fragments (enriched CRE counts) by the genomic integration frequency of each CRE (inserted CRE counts). This normalization accounts for differences in lentiviral integration efficiency across CREs.

technologies have identified millions of cCREs across many different cell types and tissues. However, these technologies are descriptive, as the binding of a protein or specific histone mark to a certain DNA sequence does not ultimately mean it is a functional CRE. In addition, it is hard to assess the effect of nucleotide variants in these cCREs on regulatory activity using these technologies.

Massively parallel reporter assays (MPRA) overcome these hurdles, allowing to test the activity of thousands of sequences and variants within them for their regulatory activity by measuring a transcribed barcode[7]. We previously developed a lentivirus-based MPRA (lentiMPRA), where cCREs are integrated into the host genome and can be tested in a wide-array of cell types[8]. By testing a similar episomal MPRA library side-by-side, we showed that this 'in genome' readout is more strongly correlated with ENCODE annotations and sequence-based models and later work on numerous MPRA technologies showed that it also provides higher cell-type specificity predictions than episomal MPRA[9]. However, while MPRA can reveal the regulatory activity of a sequence, it cannot identify the proteins binding to that sequence nor its epigenetic modifications.

Here, we develop a technology, e2MPRA, by combining lentiMPRA with CUT&Tag and ATAC-seq (Fig. 1). We used e2MPRA to analyze synthetic enhancers containing liver-specific transcription factors and dissect their epigenetic roles. Additionally, we examined a perturbation library of nine enhancers containing the POU::SOX motif, which is essential for pluripotency, demonstrating the applicability of this technology for identifying key motifs involved in epigenetic function. This technology can systematically characterize thousands of cCREs and their variants for their functional effect on regulatory activity and epigenetic modification, allowing the dissection of the DNA and epigenetic regulatory code side by side.

## Results
### e2MPRA development
To develop e2MPRA, we first designed a pilot library consisting of 400 elements, each 100 base pairs (bp) in length (Methods), which have

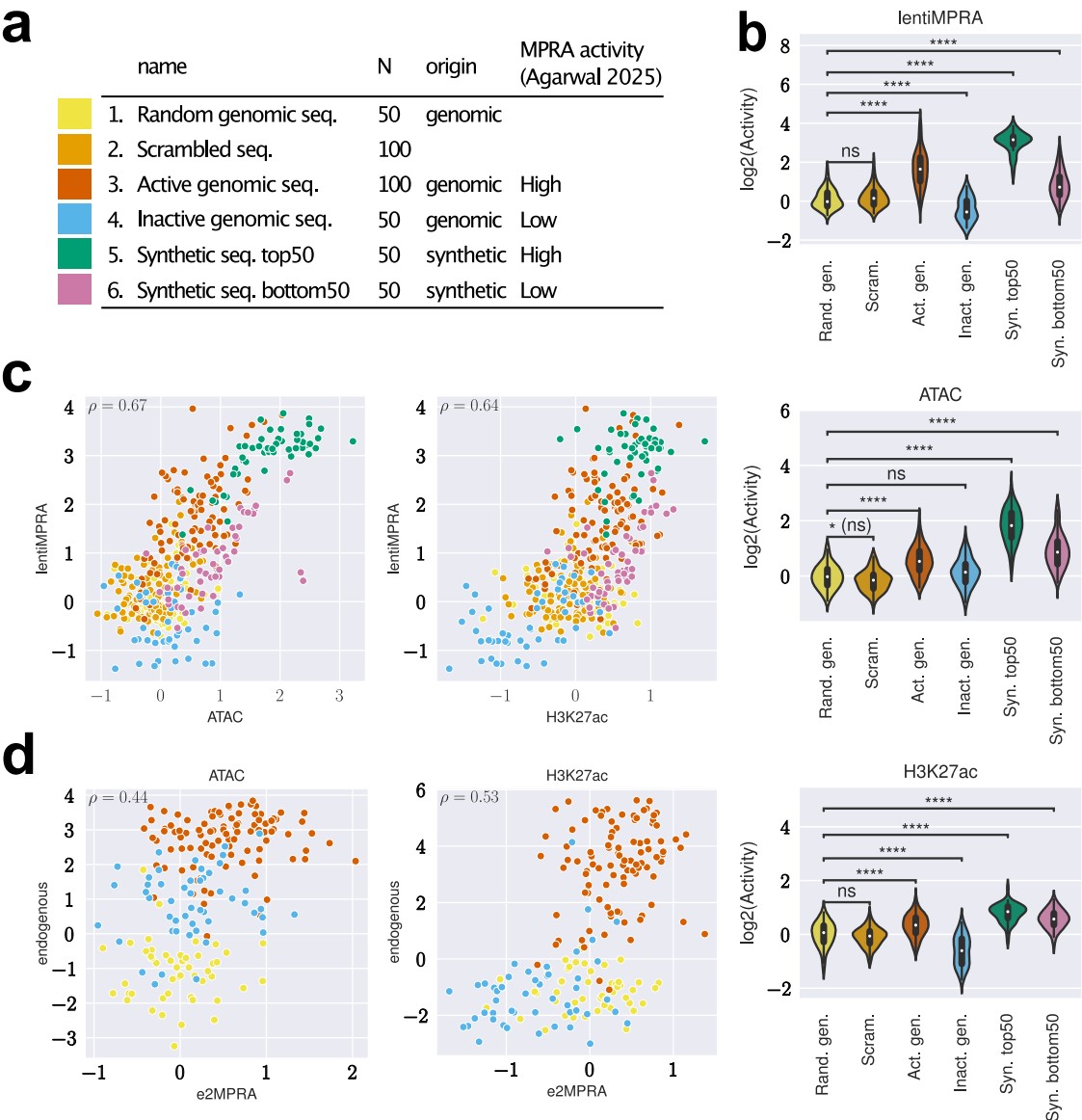

**Fig. 2 | Properties of the pilot library. a** Pilot library composition. **b** Violin plots showing the distribution of log$_2$-transformed epigenetic activities (log2(Activity)) of the pilot library elements, measured by lentiMPRA, ATAC-seq and H3K27ac CUT&Tag, across six categories: random genomic sequences (yellow), scrambled sequences (green), active genomic sequences (red), inactive genomic sequences (blue), top 50 synthetic sequences (green), and bottom 50 synthetic sequences (purple). To ensure comparability of activity scores across different assays, replicates, and libraries, trimmed mean of M-values (TMM) normalization for each replicate to define the expectation of log2(Activity) of random genomic sequences as 0. Statistical significance was determined by two-sided Mann–Whitney–Wilcoxon tests with Benjamini–Hochberg correction, comparing each category to random genomic sequences as a negative control. (****: adj-$P \leq 1.0e-4$; ns not significant; *(ns): $1.0e-2 < P <= 5.0e-2$, but adj-$P > 5.0e-2$). In the box plots (black), the median is indicated by the white point; the lower and upper bounds of the box represent the first (25th percentile) and third (75th percentile) quartiles, respectively; and the lower and upper whiskers indicate the minimum and maximum values within 1.5× the interquartile range (IQR). **c** Scatter plots comparing log$_2$(Activity) of ATAC-seq or H3K27ac CUT&Tag (x-axis) against lentiMPRA (y-axis). Each dot is colored by its category as shown in panel (**a**). Spearman's correlation coefficients (ρ) are indicated in each plot. **d** Scatter plots comparing log$_2$(Activity) measured by ATAC-seq or H3K27ac CUT&Tag versus normalized mapped read counts from endogenous genomic regions. Each dot represents a genomic CRE from the library and is colored by its category as shown in panel (**a**). Spearman's correlation (ρ) values are shown. Source data are provided as a Source Data file.

been previously tested for regulatory activity using lentiMPRA in HepG2 cells[10,11]. The pilot library included six distinct sequence categories: 1) 50 random genomic sequences as negative controls; 2) 100 scrambled sequences as negative controls; 3) 100 active CREs from the human genome; 4) 50 inactive genomic sequences; 5) the top 50 and 6) the bottom 50 synthetic sequences by MPRA activity from our previous paper (Fig. 2a). These sequences were synthesized, amplified, and coupled with a random 15-bp barcode before insertion into a lentiMPRA plasmid containing an EGFP reporter vector to generate a sequence-barcode library (Fig. 1a, Supplementary Fig. 1). The resulting

pilot library was packaged into lentivirus and transduced into HepG2 cells at a multiplicity of infection (MOI) of 50 (Fig. 1b). To minimize background signal arising from unintegrated lentiviral DNA, infected cells were passaged twice and cultured for 10 days following infection. Using these cultured cells, we simultaneously performed lentiMPRA, ATAC-seq and CUT&Tag specifically on the library to quantify transcriptional activity and epigenetic modifications in parallel (Fig. 1c, Methods). We conducted a single round of viral infection followed by three independent replicate experiments per assay, which served as technical replicates for downstream analyses.

We first analyzed the regulatory function of each sequence in the library using MPRAflow[12]. The regulatory activity of each element was quantified as the ratio of RNA barcode counts to the corresponding DNA barcode counts. Since barcode sequences are randomly associated with each element, we included only barcodes that were observed in both DNA and RNA barcode datasets within each replicate. Additionally, elements with fewer than 5 unique barcodes were excluded from the analysis. After filtering, the final dataset contained on average 79.1 unique barcodes per element, with only one element excluded (Supplementary Fig. 2a). This ensured a robust evaluation of activity and helped correct for site of integration biases. We observed a high correlation for the number of UMIs per barcodes (Spearman's $\rho = 0.78-0.89$; Supplementary Fig. 2b) and RNA/DNA ratio (Spearman's $\rho > 0.98$; Supplementary Fig. 2c) across the three replicates. As expected, active genomic CREs and top 50 synthetic sequences exhibited significantly higher regulatory activity compared to random genomic sequences and scrambled sequences (Fig. 2b). Inactive genomic and bottom 50 synthetic sequences showed low activity, recapitulating previous results.

Next, we investigated epigenetic activities of CREs in the pilot library by applying ATAC-seq and H3K27ac CUT&Tag, both well-characterized enhancer marks. In these assays, Tn5-tagmented elements were enriched using primers that specifically amplify sequences in the pilot library and simultaneously labeled with unique molecular identifiers (UMIs) (enriched CRE count; Fig. 1d, e). To estimate the genomic integration frequency for each element, we quantified "non-tagmented" elements from the input genomic DNA (gDNA) using the same primers (inserted CRE count; Fig. 1d, e). We observed an average of 95.9 and 68.8 UMIs per CRE in ATAC-seq and H3K27ac CUT&Tag, respectively, while the inserted CRE count had an average of 306.7 UMIs per CRE (Supplementary Fig. 2d). The enriched CRE counts or inserted CRE counts were highly correlated between replicates (Spearman's $\rho = 0.83-0.97$; Supplementary Fig. 2e). The epigenetic activity of each CRE was quantified as the ratio of enriched CRE count to inserted CRE count (Fig. 1e). Though we obtained slightly better correlation in UMI counts across replicates for ATAC-seq and H3K27ac CUT&Tag (Supplementary Fig. 2e) compared to the lentiMPRA result (Supplementary Fig. 2b), we observed a lower moderate log2(Activity) correlation between replicates (ATAC-seq, Spearman's $\rho = 0.64-0.66$; H3K27ac CUT&Tag, Spearman's $\rho = 0.48-0.53$, Supplementary Fig. 2f) than that observed for lentiMPRA (Supplementary Fig. 2c). These moderate correlations are consistent with the way that the present method directly counts CRE fragments using UMIs, leading to greater error propagation when calculating activity, whereas lentiMPRA uses the average of multiple barcodes. To mitigate noise, we summed the counts across replicates and then computed activity by dividing the total enriched count by the total inserted count for each element.

We found that active genomic and top 50 synthetic sequences exhibited significantly higher activity than random genomic sequences and scrambled sequences (Mann–Whitney–Wilcoxon tests with Benjamini–Hochberg correction, adj-$P$ values < 0.0001), consistent with the lentiMPRA result (Fig. 2b) with ATAC-seq and H3K27ac CUT&Tag activities across the library correlating with lentiMPRA activity (Spearman's $\rho = 0.67$ and 0.64, respectively; Fig. 2c). As expected, the bottom 50 synthetic sequences showed lower lentiMPRA activity than active genomic sequences; however, they displayed relatively high ATAC-seq and H3K27ac signals. This observation is consistent with the fact that all synthetic sequences were composed of active TF motifs, and that the bottom 50 sequences are enriched for binding sites of pioneer factors (e.g., ONECUT1, PPARA), rather than transcriptional activators (e.g., HNF1A, XBP1) (Supplementary Fig. 2g).

We observed a moderate correlation between epigenetic activity as measured by e2MPRA and endogenous ATAC-seq and H3K27ac ChIP-seq signals obtained from the ENCODE dataset[13] (ATAC, Spearman's $\rho = 0.44$; H3K27ac, Spearman's $\rho = 0.53$; Fig. 2d). This could be due to a technical aspect with e2MPRA where specific primers are used to count how many times each element has been tagmented, while genome-wide Cut&Tag and ATAC-seq measure the enrichment based on the number of unique reads in each region. Taken together, our pilot library suggests that although e2MPRA does not quantitatively measure epigenetic activity in a comparable manner to the endogenous genomic context, it can qualitatively capture epigenetic modifications to assess intrinsic regulatory potential.

**Dissecting TFs function on epigenetic modifications via e2MPRA**

Next, we set out to use e2MPRA to systematically evaluate the effect of TFs on epigenetic modifications. We designed a library of synthetic cCREs that contain nine binding motifs of TFs that are known to be expressed in the liver: CEBPA, CTCF, FOXA1, HNF1A, NR2F2, ONECUT1, PPARA, REST, and XBP1[11,14] (Fig. 3a). These TF motifs were systematically arranged into three classes on two distinct neutral templates (chr9:83712634-83712733 and chr2:211153273-211153372; hg19) previously shown not to have enhancer activity in HepG2 cells[11,14]. Class 1 ($N = 27$ per template, total $N = 54$) consisted of homotypic arrangements of 1, 2, or 4 evenly spaced copies of each TF motif, to assess the impact of the number of motifs on their epigenetic activity. Class 2 consisted of heterotypic arrangements of two different motifs to examine their cooperation ($N = 288$). Class 3 comprises heterotypic arrangements of four distinct motifs in all possible combinations to explore further complex interactions ($N = 6048$). We then characterized this library using e2MPRA in HepG2 cells. cCRE activities on the two templates were treated as replicates, yielding two replicates per cCRE, in addition to the three biological replicates of independent library infections.

For the lentiMPRA assay, we observed an average of 157.4 unique barcodes per cCRE (Supplementary Fig. 3a), and the correlations between replicates were high (Spearman's $\rho \approx 0.9$; Supplementary Fig. 3b). For the e2MPRA, we detected on average 406.0 UMIs per cCRE for inserted counts, and 66.5 and 38.8 UMIs per cCRE for enriched counts in ATAC-seq and H3K27ac CUT&Tag, respectively (Supplementary Fig. 3c). The correlations of the activity between replicates for ATAC-seq and H3K27ac were moderate, ranging from 0.4 to 0.5 (Supplementary Fig. 3d), as the e2MPRA data exhibited greater variability compared to lentiMPRA, consistent with our pilot library results. Additionally, sequences containing the NR2F2 motif were rarely amplified or not detected in genomic DNA, likely due to its poly C-like sequence (Supplementary Data 11; see Methods). Therefore, we excluded these data from subsequent analyses, and the remaining eight TF motifs were used for further analyses.

For Class 1, we found that having more CEBPA, FOXA1, HNF1A and XBP1 motifs significantly increased regulatory activity (MPRA), suggesting their primary role as direct transcriptional activators, as shown previously[11] (t-test with Benjamini–Hochberg correction, FDR < 0.05; Fig. 3b, Supplementary Fig. 4). We found that HNF1A and ONECUT1 motifs were associated with an increase in ATAC signal, fitting with their characterized role as pioneer TFs[15,16]. The PPARA motif was shown to affect both chromatin accessibility and H3K27ac modification but not transcriptional activation, i.e. having more copies of it did not lead to regulatory activity, consistent with its interaction with chromatin remodeling factors[17]. Conversely, REST and CTCF, which are known to function as repressors in a context-dependent manner[18–20], showed no significant correlation with epigenetic activity (Supplementary Fig. 4).

Next, we compared Class1 (homotypic arrangements of individual TF motifs) and Class 2 (heterotypic arrangements of two different motifs) to examine synergistic effects of TF motif pairs. For each pair, we performed linear regression to quantify the individual contributions to epigenetic activity, introducing a synergistic interaction term. We observed multiple cooperative effects of two TF motifs on the transcriptional activity measured by lentiMPRA (Fig. 3c, d, Supplementary Fig. 5a), but not for ATAC-seq and H2K27ac activities

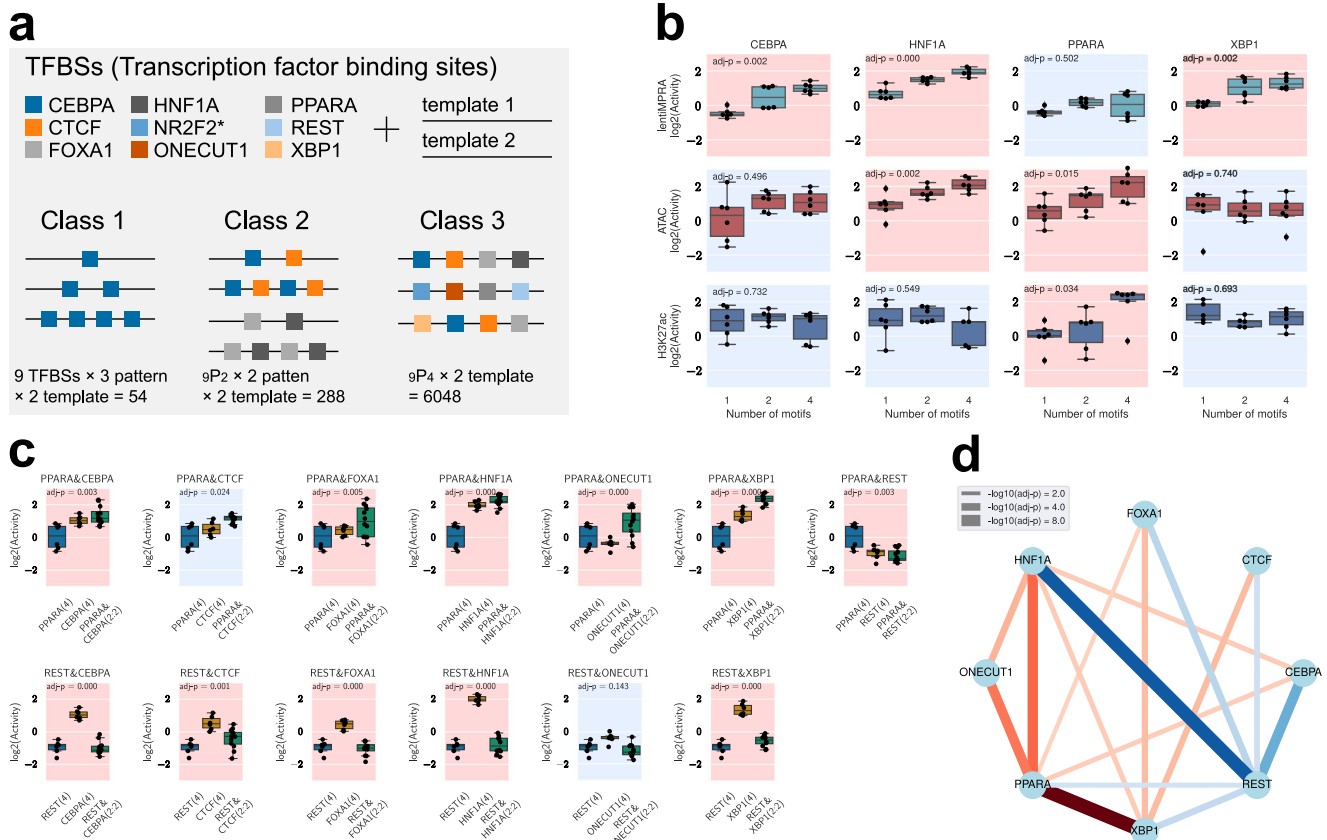

**Fig. 3 | Synthetic cCRE library dissecting TF motif effect on regulatory activity and epigenetic modifications. a** Synthetic cCRE library design. Nine TF motifs were systematically arranged on two neutral templates into three distinct classes (Class 1: homotypic, Class 2: two-TF combinations, and Class 3: four-TF combinations) to evaluate epigenetic function. (*NR2F2 containing sequences were not sufficiently detected in the assays and excluded from subsequent analyses). **b** Regulatory and epigenetic activities (lentiMPRA, ATAC-seq, and H3K27ac CUT&Tag) measured for Class 1 sequences. Significant Spearman's correlations (FDR < 0.05) between TFBS copy number (x-axis) and log2(Activity) (y-axis) are indicated with a red background, and non-significant correlations are indicated in blue. Two-sided Spearman rank correlation tests were performed, and $p$-values were adjusted using the Benjamini–Hochberg correction. $n = 2$ templates × 3 replicates per box plot. **c** Representative examples showing synergistic transcriptional effects when combining PPARA as well as the repressive effect of REST motifs

over other TF motifs (Class 2). Transcriptional activities of homotypic arrangements (Class 1; four identical motifs) were compared to heterotypic arrangements (Class 2; two motifs combined in a 2:2 ratio). Statistically significant synergistic effects, as determined by linear regression and two-sided $t$-tests with Benjamini–Hochberg correction (FDR < 0.01), are indicated by a red background and non-significant combinations in blue. $n = 2$ templates × 3 replicates per box plot. **d** Network visualization of significantly synergistic TF pairs (FDR < 0.01). Red lines indicate positive synergy, and blue lines indicate repressions. Line thickness corresponds to the $-\log_{10}$(adjusted $p$-value) for the significance of the interaction term. In the box plots, the median is indicated by the black line; the lower and upper bounds of the box represent the first (25th percentile) and third (75th percentile) quartiles, respectively; and the lower and upper whiskers indicate the minimum and maximum values within 1.5× the interquartile range (IQR). Source data are provided as a Source Data file.

(Supplementary Fig. 5b, c). For example, PPARA synergistically increased transcriptional activity in combination with CEBPA, CTCF, FOXA1, ONECUT1, and XBP1 (Fig. 3c upper panel), despite its homotypic arrangements increasing only ATAC-seq and H3K27ac activity but not transcriptional activity (Fig. 3b). This suggests that PPARA facilitates transcriptional activation by mediating its epigenetic activity. In addition to the synergistic effects, we also observed repressive effects mediated by REST. REST significantly reduced lentiMPRA activity when combined with any other TF except ONECUT1 (Fig. 3c, d). This suggests that REST represses transcription in the presence of adjacent activators.

To further investigate regulatory grammar, we analyzed Class 3 sequences (heterotypic arrangements of four distinct motifs in all possible combinations) and assessed the effects of motif order on regulatory activity. For each combination of four motifs ($N = 70$), we performed one-way ANOVA to determine the significance of order on regulatory activity. Among the significant combinations affecting transcriptional activity, the sequence containing four TF motifs [HNF1A, PPARA, REST, XBP1] showed the largest differential regulatory activity depending on TF motif order (Fig. 4a, b). Many of these TF

motif combinations showed significant effects on transcriptional activity, but not ATAC-seq or H3K27ac activities, which is consistent with epigenetic modifications occurring in broader regions around cCREs (FDR < 0.01; Fig. 4b). In addition, we further investigated positional enrichment of each TF motif in the top 200 and bottom 200 sequences, with position 1-to-4 being distal-to-proximal from the minimal promoter (mP) (Fig. 4c). Specifically, transcriptional activators such as HNF1A and XBP1 showed greater enhancement at positions closer to the mP in the top 200 sequences and depletion in the bottom 200. Conversely, the REST repressor exhibited stronger repressive effect when located closer to the mP. Taken together, these results demonstrate that e2MPRA allows the analysis of epigenetic activity of TFs and their cooperative function independent of their transcriptional activity.

**e2MPRA analysis of variant effects**

We next set out to test the ability of e2MPRA to simultaneously characterize the effect of variants on regulatory activity and epigenetic marks. We selected five sequences positive for enhancer activity from a previous lentiMPRA[10] carried out in induced pluripotent stem cells

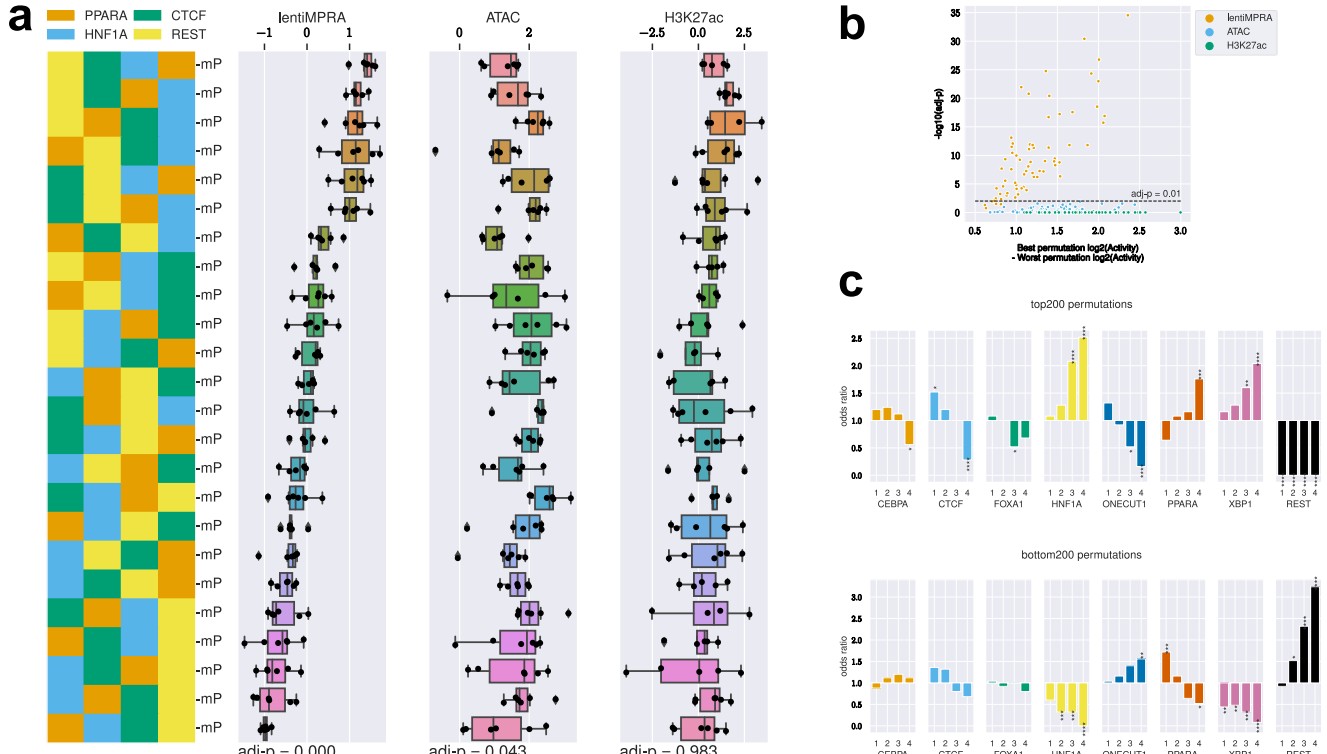

**Fig. 4 | TF motif order affects enhancer activity. a** Representative example demonstrating how rearranging motif order (HNF1A, PPARA, REST, and XBP1) affects enhancer epigenetic activity. $n = 2$ templates × 3 replicates per box plot. **b** Statistical analysis (one-way ANOVA) with Benjamini–Hochberg correction evaluating whether changes in motif order within each set of four distinct motifs (24 permutations per set) significantly affects epigenetic activities. All 70 unique combinations (selecting 4 TF motifs from 8, $_8C_4 = 70$) were tested. The x-axis represents the difference in activity between the permutation with the highest activity and lowest activity among the 24 permutations tested for each combination. The y-axis indicates the corresponding statistical significance ($-\log_{10}$ adjusted *P*-value). **c** Positional enrichment analysis of TF motifs within the top and bottom 200 permutations ranked by transcriptional activity. Statistical significance of enrichment was determined by hypergeometric tests with Benjamini–Hochberg correction (*: adj-$P < 5.0e{-}2$; **: adj-$P < 1.0e{-}2$; ***:adj-$P < 1.0e{-}3$; ****:adj-$P < 1.0e{-}4$). Motif positions are numbered from distal (position 1) to proximal (position 4) relative to the minimal promoter. In the box plots, the median is indicated by the black line; the lower and upper bounds of the box represent the first (25th percentile) and third (75th percentile) quartiles, respectively; and the lower and upper whiskers indicate the minimum and maximum values within 1.5× the interquartile range (IQR). Source data are provided as a Source Data file.

(iPSCs; WTC11 line). These sequences had ATAC-seq and ChIP-seq peaks for POU5F1/SOX2 in human ESCs and iPSCs, as well as being in close proximity (<1 Mb) to genes associated with pluripotency and/or early development (Fig. 5a, Methods). We generated a single-nucleotide substitution library (Fig. 5b), where each nucleotide within the 100-bp sequence was individually mutated to each of the three alternative nucleotides ($N = 300$ variants per CRE). We then performed e2MPRA using WTC11 iPSCs to quantify the impact of these variants on transcriptional and epigenetic activities.

For the lentiMPRA, we observed an average of 211.8 unique barcodes per CRE (Supplementary Fig. 6a), and the activity of variants was highly correlated between replicates (Spearman's $\rho \approx 0.95$; Supplementary Fig. 6b). For the ATAC-seq and CUT&Tag H3K27ac, we detected on average 754.4 UMIs per element for inserted counts, and 43.3 and 56.5 UMIs per element for enriched counts, respectively (Supplementary Fig. 6c). The correlations between replicates were lower for both ATAC-seq and H3K27ac assays (Spearman's $\rho \approx 0.4$–$0.5$; Supplementary Fig. 6d), consistent with previous observations from the pilot and HepG2 libraries. One of the five active CREs (seq6846_R; Fig. 5a) yielded insufficient read coverage and was therefore excluded from downstream analyses. The remaining four CREs showed high lentiMPRA activities ($\log_2(\text{Activity}) = 0.66$–$4.79$) for the wild type sequence fitting with their selection as active enhancers (Fig. 5a). Thus, we focused our subsequent analyses on these four CREs. For the single-nucleotide substitution libraries, we quantified the effects of each variant using a linear regression model and found positional clusters of

variant effects (Fig. 5c, d, Supplementary Fig. 7). For example, CRE seq68781_R, located near the *FOXK1* gene, contains ETS and POU5F1::SOX2 motifs, and their mutations significantly decreased its activity (Fig. 5c).

Analyzing ATAC-seq and H3K27ac activities, we observed noisy signals and no clusters of variant effects around putative TF motifs in any CREs (Fig. 5c, d, Supplementary Fig. 7). This could be due to e2MPRA directly using read counts of amplified variants without using barcode counts, thus having weaker statistical power. To overcome this, we introduced 6 bp randomization with 1bp-sliding windows for each CRE (two randomizations per window, total $N = 190$ per CRE; Fig. 5b). To quantify the impact of these perturbations, we calculated the perturbation effect at each position (k) as the median epigenetic activity across all sequences affected by perturbation at position k, then computed the median absolute deviation (MAD) score using wild-type (WT) enhancer activity as the baseline (set to 0). Finally, we applied a modified Canny edge detection algorithm to the position vs. MAD score data, smoothing and identifying regions with sharp epigenetic activity changes upon perturbation as significant functional peaks (Methods). This approach found that disruptions of POU5F1::-SOX2 and ETS motifs in the seq68781_R (Fig. 5e) significantly decreased ATAC-seq and H3K27ac activities, consistent with the single-nucleotide mutation results (Fig. 5c). In the previously characterized *POU5F1* enhancer[21] (POU5F1_DE_core), we found that disruption of POU5F1::SOX2 motif decreased both regulatory activity and ATAC signals (Fig. 5f). In addition, while the disruption of the YY1 motif

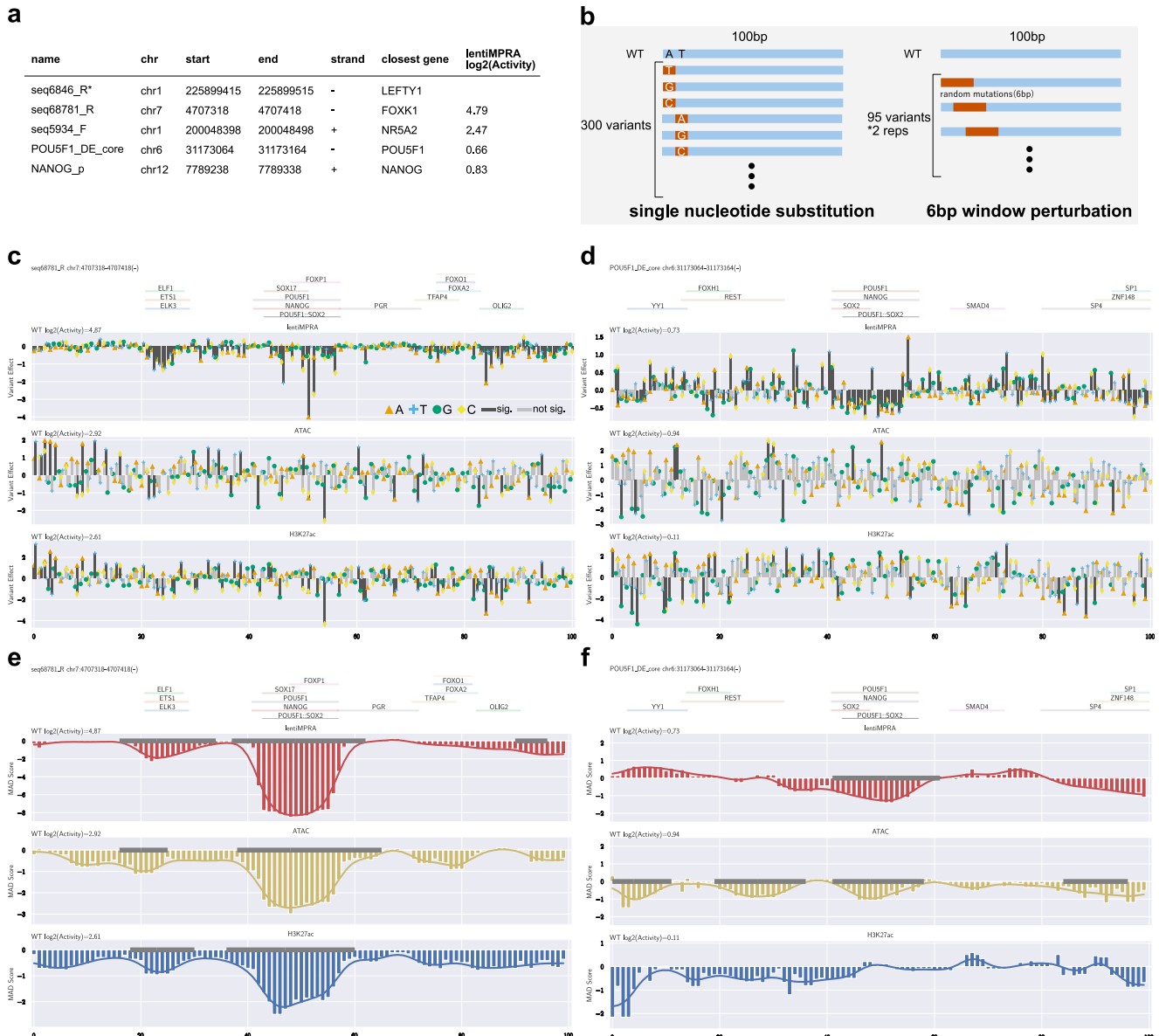

**Fig. 5 | CRE perturbation analyses. a** CREs selected for perturbation analysis. Variants derived from seq6846_R (*) were not sufficiently detected in the assays and were therefore excluded from subsequent analyses. **b** Two strategies were used for enhancer perturbation analyses. In the single-nucleotide substitution approach (left), each nucleotide in a 100-bp CRE was individually mutated to all three alternative nucleotides (300 variants per CRE). In the 6-bp window perturbation approach (right), consecutive 6-bp segments were randomly mutated across the enhancer sequence using a sliding window shifted by 1 bp increments, resulting in 95 variants per CRE replicate. This randomization was performed twice independently, generating a total of 190 variants per CRE. Analysis of single-nucleotide substitution effects on epigenetic activities (lentiMPRA, ATAC, and H3K27ac CUT&Tag) for seq68781_R (**c**) and POU5F1_DE_core (**d**). Annotated TF motifs[49] are shown above each plot. Bar colors represent statistical significance (black: significant at $P < 0.01$; gray: not significant). *P* values were obtained from multiple linear regression. Analysis of 6-bp window perturbation effects for seq68781_R (**e**) and POU5F1_DE_core (**f**). Positional effects of mutations were quantified as median absolute deviation (MAD) scores at each nucleotide position and smoothed using a Gaussian filter (line plot). Regions significantly affected by mutations, as identified by edge detection, are indicated by gray lines along the baseline. Source data are provided as a Source Data file.

increased regulatory activity, it reduced ATAC activity (no significance for H3K27ac), suggesting that the transcriptional repression by YY1 is associated with chromatin opening mediated by its own binding. It was previously shown that both POU5F1::SOX2 and YY1 motifs are conserved among species and required for enhancer activity[21]. Overall, our results demonstrate that e2MPRA effectively enables high-resolution identification of functional motifs, providing a powerful framework for systematic dissection of regulatory logic.

## Discussion

In this study, we used lentiMPRA both for testing the function of cCREs and for the enrichment of specific cCREs within the genome (i.e.

allowing for many copies of a specific cCRE to be integrated into the genome). In-genome enrichment of these cCREs enabled us to use CUT&Tag and ATAC-seq to characterize the proteins that bind to these sequences and their epigenetic and nucleosome profile. This approach allowed us to: 1) develop a technology that tests regulatory sequences for their function coupled with TF binding and epigenetic modification; 2) generate a catalog of regulatory elements that are characterized not only by their transcriptional activity but also via open chromatin and epigenetic modifications; 3) obtain a better understanding of regulatory grammar and how its alteration can lead to phenotypic consequences.

Our pilot library showcased the feasibility of this approach. While we observed a moderate correlation between replicates for ATAC-seq

and H3K27ac CUT&Tag (Supplementary Fig. 2f) which is due to the use of UMI counts, summing up these counts across replicates showed the expected results for negative and positive controls. They also showed a good correlation with our lentiMPRA results (0.64–0.67; Fig. 2c). Factors that could drive differences are the way each method measures activity, i.e. lentiMPRA using an average of multiple barcodes, whereas ATAC-seq and CUT&Tag in e2MPRA quantifies Tn5 tagmentation events using UMIs.

It is generally thought that the combinatorial signal of multiple epigenetic modifications contributes to enhancer activity more than single epigenetic modifications alone[22]. Indeed, our previous work integrating multiple biochemical features at CREs (TF ChIP-seq, histone ChIP-seq, DNase-seq, and ATAC-seq) enabled to predict lentiMPRA activity to a certain extent; the Pearson correlation was around 0.7–0.75 (Fig. 3b in Agarwal et al. Nature 2025)[10], indicating that epigenomic features alone have limited explanatory power. These findings further emphasize the value of our approach to analyze multiple epigenetic modifications separately.

We further observed a low correlation between e2MPRA and endogenous epigenetic signals. This could be due to differences in measurement of the two methods, i.e. e2MPRA activities were detected by specific primers and measured in the vector context, while endogenous activities were detected by random tagmentation. The use of specific primers in e2MPRA allows for unbiased comparisons between sequence variants within the library. Therefore, e2MPRA could be more suitable for the parallel quantitative comparison of enhancer activities among sequence variants rather than for recapitulating endogenous epigenetic modifications.

We applied e2MPRA to evaluate synthetic enhancers containing defined TF motifs and demonstrated its effectiveness in measuring epigenetic activity. We showed that CEBPA, FOXA1, HNF1A, and XBP1 motifs increased transcriptional activity, consistent with previous results[11,14] and their known roles as transcriptional activators. In addition, HNF1A and ONECUT1, previously characterized as pioneer factors[16], were shown to increase chromatin accessibility, further validating the ability of e2MPRA to disentangle epigenetic regulation from transcriptional activation, which conventional MPRAs alone cannot achieve. Moreover, e2MPRA enabled the detection of changes in chromatin accessibility and H3K27ac modification mediated by PPARA, likely through its interactions with chromatin remodeling factors. The ability to dissect distinct molecular functions of individual TFs underscores the utility of e2MPRA in elucidating the regulatory grammar of gene expression.

While CEBPA, FOXA1, and XBP1 increase lentiMPRA activity along with the number of their motifs, we did not observe epigenetic changes for them (Fig. 3b, Supplementary Fig. 4). This could be due to these factors being responsible for activating gene transcription by direct interaction with gene promoters without epigenetic modification and/or not having a role related to H3K27ac. Another reason could be due to the limitation of e2MPRA technology, requiring additional reads and replicates to obtain more robust epigenetic results. Thus, we cannot exclude the possibility that these factors are involved in epigenetic modifications.

The pioneering activity of FOXA1 is still being characterized. While early studies proposed that FOXA1 acts as a pioneer factor capable of binding to closed chromatin and facilitating subsequent transcriptional activation[22], more recent reports have suggested that its pioneering capacity may be limited to a subset of strong binding sites or may depend on cellular context[23]. In our e2MPRA assay, FOXA1 increased transcriptional activity without a corresponding increase in chromatin accessibility or H3K27ac modification (adjusted $p = 0.115$ in ATAC-seq; Supplementary Fig. 4), suggesting that under our experimental conditions, FOXA1 primarily functioned as a transcriptional activator rather than as a pioneer factor. Nevertheless, given the moderate statistical significance of the observed effects, we cannot rule out a pioneering activity for FOXA1.

REST is a well-characterized transcriptional repressor that mediates silencing through recruitment of chromatin modifiers, including HDACs, LSD1, G9a/GLP, and PRC1/2[24,25]. In our assay, REST repressed transcription when present together with transcriptional activators (Fig. 3c), suggesting that it exerted the strongest effect among the factors tested alongside it. By contrast, we did not detect a significant depletion of H3K27ac and closed chromatin for REST (Supplementary Fig. 4). This could be due to a variety of factors, including noisy measurements in this specific e2MPRA. Further technical improvement may be required to better understand its molecular function.

We further demonstrate that e2MPRA is a powerful tool to characterize the effect of variants on epigenetic modifications. Analysis of four regions that interact with POU5F1 and SOX2 showed that mutations in the POU5F1::SOX2 motif not only alter transcriptional activity but also impact chromatin accessibility and H3K27ac modification. This fits with pluripotent factors being known to function as pioneer factors, with SOX2 in particular being reported to interact with the histone acetyltransferase p300[26]. In addition, e2MPRA allowed the identification of motifs responsible for epigenetic function in enhancers, e.g. the YY1 motif in the POU5F1 distal enhancer, which requires chromatin accessibility but negatively regulates transcription. The "CR4-C" region[27] in this enhancer contains the YY1 motif and is required for POU5F1 expression. It is reported that YY1 directly binds to this region (ENCODE, ENCFF509GYP) in human ESCs, and plays a role in regulating pluripotency by directly interacting with OCT4 and BAF chromatin remodeling factors in mouse ESCs[28].

It is worth noting that e2MPRA also has several caveats and technical limitations. First, as cCREs are placed upstream of a promoter in the vector, it lacks the endogenous genomic context, such as enhancer–promoter looping and chromatin 3D architecture. Second, e2MPRA uses an artificial minimal promoter instead of the cognate promoter. Therefore, e2MPRA cannot detect epigenetic functions that are associated with 3D chromatin architecture or specific promoter-enhancer compatibilities. Due to the detection sensitivity, e2MPRA currently requires shorter CREs (~100 bp), making it unsuitable for the analysis of longer elements such as super-enhancers. In addition, as this was a proof-of-concept study, we were more conservative and used a small number of sequences in these assays; however, this technology could be scaled up by increasing library size, cell numbers and number of viral integrations per cell. Further development of this technology could overcome many of these limitations. In summary, our study demonstrates the power and feasibility of this approach, which enables high-resolution characterization of regulatory activity along with epigenetic function for a large number of sequences, providing valuable insights for discovering functional elements and identifying disease-associated variants.

## Methods

### MPRA library design

**HepG2 pilot library.** To design the pilot library, we selected sequence features listed in Fig. 2a. 100 active CREs were selected from the top 100 sequences in our previous lentiMPRA in HepG2[10], as well as the overlap with H3K27ac peaks (ENCODE data: ENCFF001SWK). 50 inactive genomic sequences were selected as the bottom 50 sequences. For synthetic enhancers, we selected the top 50 and bottom 50 sequences based on MPRA signals from the library used in Smith et al.[11], which consists of combinations of 12 active TF motifs. 50 random genomic sequences were selected from genomic sequences that do not overlap without H3K27ac, H3K27me3 marks, and RepeatMasker annotation. Scramble sequences were generated by randomizing nucleotides using 100 active genomic sequences. Each CRE was 200 bp in length, and flanked by 15 bp adapter sequences (5'-AGGACCGGAT-CAACT and CATTGCGTGAACCGA-3').

To generate 100 bp and 150 bp libraries, the initial 200 bp CRE sequences were cropped from the center of each sequence and added

another adapter sequences (5′-AATGCTAGCGCATGG and CTGCAACCTACGGAA-3′). We tested all the 100 bp, 150 bp and 200 bp libraries with e2MPRA. However, we were not able to effectively obtain PCR amplification followed by ATAC or CUT&Tag for the 150 bp and 200 bp libraries. Therefore, we decided to use the 100 bp library in the following experiments.

**HepG2 synthetic enhancer library.** Seven TF binding motifs (CEBPA, FOXA1, HNF1A, NR2F2, ONECUT1, PPARA, and XBP1) used in Smith et al.[11] were selected based on their correlation significance with transcriptional activity. In addition, we included the CTCF and REST sites used in Georgakopoulos-Soares et al.[14], resulting in a total of nine TF motifs in this library. These motifs were arranged in various combinations (Class 1, 2, 3; Fig. 3a) on two neutral DNA templates (hg19 chr9:83712634-83712733 and chr2:211153273-211153372), which were the same as those used in the previous study but cropped 100 bp from the center. In addition to these synthetic enhancer sequences, we also included 200 control sequences selected from four sequence features of the pilot library based on lentiMPRA activity: random genomic sequences, scrambled sequences, and active/inactive genomic sequences (50 each). These control sequences were used only for TMM normalization.

**WTC11 enhancer perturbation library.** To select enhancer candidates for perturbation in the WTC11 iPSC line, we first collected POU5F1 and SOX2 peaks from TF ChIP-seq data and H3K27ac ChIP-seq peaks from ChIP-Atlas[29] (last accessed: 2023-05-11). These peaks were overlapped with functional WTC11 CREs identified previously[10], and the intersected CREs were scanned for POU5F1:SOX2 binding motifs using FIMO[30] ver. 5.5.1, resulting in 1000 candidates. Among these candidates, we selected five active (seq6846_R, seq68781_R, seq5934_F, POU5F1_DE_core, NANOG_p) located near genes of interest within 1 Mb and four inactive sequences (seq16767_R, seq34039_F, seq34899_R, seq2846_R). The inactive sequences were included to identify potential variants that relieve repressive elements and lead to their activation. However, their MPRA activity scores were too weak and therefore excluded from subsequent analyses (Supplementary Data 8). We trimmed these sequences to 100 bp, ensuring that the POU5F1:SOX2 motif is included. To generate the single nucleotide substituted subset, we created sequences in which each nucleotide at every position of the target sequence was replaced with each of the other three nucleotides. To generate the 6-bp window perturbation subset, we created sequences in which random mutations were introduced within 6-bp windows, shifting by 1 bp at a time. To prevent the emergence of TF motifs due to random mutations, we extracted a 6-bp region upstream and downstream of the mutated site (totaling 18 bp) and matched with known binding motifs using FIMO. If known motifs were detected, the introduced mutation was rejected and a new mutation process was performed recursively. The mutation process was repeated twice, and both replicates were included in the library. We also included the same 200 control sequences from the HepG2 library which were also used only for TMM normalization.

### Generation of MPRA libraries
The MPRA libraries were generated as previously described[12], with minor modifications. In brief, all library sequences were synthesized as a Twist oligonucleotide pool with adapter sequences at both ends (adapter 02 for the pilot library and the WTC11 library; adapter 03 for the HepG2 library; Supplementary Fig. 1a). The oligonucleotide pool was amplified by two rounds of PCR using NEBNext High-Fidelity 2X PCR Master Mix (NEB; Supplementary Fig. 1b). The first-round PCR was performed for 10 cycles using the primer set 5BC-AG02/03-f01 and 5BC-AG02/03-r01 (Supplementary Data 11) to attach a minimal promoter. The second-round PCR was performed for 10 cycles using the primer set 5BC-AG-f02 and 5BC-AG-r02 (Supplementary Data 11) to

attach 15-bp random barcodes for lentiMPRA. For both PCR reactions, the following cycling program was used: 98 °C for 2 min; 10 cycles of 98 °C for 15 s, 60 °C for 20 s, and 72 °C for 30 s; followed by a final extension at 72 °C for 5 min. The amplified fragments were then inserted into the *Sbf*I/*Age*I site of the pLS-SceI vector (Addgene, #137725) using the NEBuilder HiFi DNA Assembly mix (NEB). The recombinant products were transformed into electrocompetent cells (NEB) following the manufacturer's protocol and incubated overnight at 37 °C on 15-cm LB agar plates with 100 µL of 100 mg/mL carbenicillin (Nacalai). For each library, we collected ~80,000 colonies for the pilot library, 1.3 million colonies for the HepG2 synthetic enhancer library, and 0.9 million colonies for the WTC11 enhancer perturbation library, aiming to obtain on average 200 barcodes per CRE. To determine the association between CREs and random barcodes, CRE-mP-barcode fragments were PCR-amplified from each plasmid library pool, and P5 and P7 flowcell adapters were attached using P5-pLSmP-ass-i# and P7-pLSmP-ass-gfp (98 °C for 1 min; 15 cycles of 98 °C for 15 s, 60 °C for 20 s, and 72 °C for 3 min; 72 °C for 5 min) (Supplementary Data 11). The fragments were then sequenced with a iSeq (the pilot library) or NextSeq Mid output 300-cycle kit (the HepG2 and WTC11 library) using custom primers (Read 1: pLSmP-ass02/03-seq-R1; Index read1: pLSmP-ass-seq-ind1; Index read2: pLSmP-rand-ind2; Read 2: pLSmP-ass-seq02/03-R2, Supplementary Data 11).

### Lentivirus packaging and titration
Lentivirus packaging, titration, and infection were performed as previously described[12], with minor modifications. Lentivirus for each plasmid library was produced by co-transfecting it with the helper plasmids pMD2.G (Addgene, #12259) and psPAX2 (Addgene, #12260) into HEK293T (CRL-3216, ATCC) cells cultured in T175 flasks using the EndoFectin Lenti transfection reagent (GeneCopoeia), according to the manufacturer's protocol. At 8 h post-transfection, the culture medium was refreshed, and ViralBoost reagent (Alstem) was added. At 48 h after adding the ViralBoost reagent, the culture supernatant was collected, filtered through a 0.45-µm PES filter unit, concentrated 50-fold using the Lenti-X concentrator (Takara), and stored at 4 °C for up to three weeks. The purified lentivirus was then used for titration. For the pilot library and the HepG2 library, HepG2 cells (HB-8065, ATCC) were plated at 1 million cells per well in 6-well plates and incubated for 24 hours. Then, serial volumes (0, 4, 8, 16, 32, 64 µL) of the lentivirus were added along with 8 µg/mL Polybrene. For the WTC11 library, WTC11 cells (P59; GM25256, Coriell Institute) were plated in 6-well plates at 20% confluency and incubated for 24 h. Then, serial volumes (16, 32 µL) of the lentivirus were added along with an equal volume of ViroMag reagent (OZ Biosciences). All infected cells were cultured for three days and then washed three times with PBS. Genomic DNA was extracted using the Wizard SV Genomic DNA Purification Kit (Promega). The multiplicity of infection (MOI) was measured as the relative amount of viral DNA (WPRE region, primer set: WPRE.F and WPRE.R; backbone, primer set: BB.F and BB.R; Supplementary Data 11) to genomic DNA (intronic region of the LIPC gene, primer set: LP34.F and LP34.R, Supplementary Data 11) by qPCR using the Thunderbird SYBR qPCR Mix (Toyobo), according to the manufacturer's protocol.

### Lentiviral infections and cell cultures
For the pilot library and the HepG2 library, 4 million or 2 million cells per replicate (the pilot library: 1 replicate; the HepG2 library: 3 replicates) were seeded in 2 wells of a 6-well plate and incubated for 24 h. The cells were then infected with the lentiviral libraries along with 8 µg/mL polybrene, with an estimated MOI of 50. For the WTC11 library, 5 million cells per replicate (3 replicates) were seeded in a 10-cm dish and incubated for 24 h. The cells were then infected with 160 µL of the lentiviral library along with 80 µL of ViroMag reagent, with an estimated MOI of 5. To eliminate residual viral DNA within the cells, infected cells were passaged twice after infection and maintained a

total of 8–10 days. EGFP fluorescence was observed to confirm that the infected library remained unsilenced after passages. The cells were collected and cryopreserved at −80 °C using 100 μL of BamBanker (Nippon Genetics) per 1 million cells. These cryopreserved cells were used for downstream assays.

### gDNA/RNA extraction and DNA/RNA barcode sequencing for lentiMPRA assay

The lentiMPRA assay was performed as previously described[12], with minor modifications. In brief, the frozen cells (pilot library: 1 million cells; HepG2 library: 2.5 million cells; WTC11 library: 10 million cells per replicate) were washed twice with PBS and lysed in RLT Plus buffer (pilot library: 100 μL; HepG2 library: 200 μL; WTC11 library: 1 mL; QIAGEN). Genomic DNA and total RNA were extracted from the lysate using the AllPrep DNA/RNA Mini Kit (QIAGEN). To prevent genomic DNA contamination, the Turbo DNA-free Kit (Thermo Fisher) was used for further purification of the RNA solution. Then, the entire purified RNA was used for reverse transcription to generate cDNA using SuperScript IV (Invitrogen). To prepare the sequencing library, 1.5 μg (pilot and HepG2 libraries) or 6 μg (WTC11 library) of genomic DNA and the entire RT product were PCR-amplified for 3 cycles (primer: P5-pLSmP-5bc-i#/P7-pLSmP-ass16UMI-gfp, Supplementary Data 11; cycling program: 98 °C for 1 min; 3 cycles of 98 °C for 10 s, 60 °C for 30 s, and 72 °C for 1 min; 72 °C for 5 min) to attach P5 and P7 flow cell adapters and UMIs to the sequencing library. The amplicons were further amplified 18 cycles using the primer set P5/P7. The final product was pooled the DNA and RNA barcode library at a molar ratio of 1:3 and sequenced on a iSeq (the pilot library) or NextSeq High output 75-cycle kit (the HepG2 and WTC11 library) with custom primers (Read 1: pLSmP-ass-seq-ind1, Read 2: pLSmP-bc-seq, i7 index: pLSmP-UMI-seq, i5 index: pLSmP-5bc-seq-R2) (Supplementary Data 11).

### ATAC and CUT&Tag for e2MPRA assay

ATAC-seq[31] (Illumina Tagment DNA Enzyme and Buffer Small Kit; Illumina) and CUT&Tag (Hyperactive In-Situ ChIP Library Prep Kit for Illumina; Vazyme) were performed on frozen cells following the manufacturer's protocol with minor modifications. For both assays, we used 500,000 cells (pilot and HepG2 libraries) or 2 million cells (WTC11 library). The following procedure was performed for 500,000 cells. For 2 million cells, all reagent volumes were increased fourfold accordingly. For the ATAC assay, the cells were washed with PBS and suspended in 50 μL of lysis buffer for nuclei isolation. The isolated nuclei were then resuspended in 50 μL of transposition mix per 500,000 cells, incubated at 37 °C for 30 minutes, and purified using the MinElute Reaction Cleanup Kit (QIAGEN).

For the CUT&Tag assay, the cells were washed with 250 μL of wash buffer, then resuspended in 50 μL of wash buffer and 5 μL of ConA bead solution. After treatment with ConA beads, the cells were washed again and resuspended in 50 μL of antibody buffer, followed by the addition of 0.5 μL of primary antibody and 0.5 μL of secondary antibody. We used H3K27ac (Abcam: ab4729) as the primary antibody and goat anti-rabbit IgG (Abcam: ab6702) as the secondary antibody. The primary antibody incubation was performed overnight at 4 °C, while the secondary antibody was incubated for 2 h at room temperature. After incubation, the solution was washed and resuspended in 150 μL of the tagmentation buffer, then incubated at 37 °C for 1 hour. The tagmented DNA was extracted by stopping the reaction with 5 μL of 0.5 M EDTA, 1.5 μL of 10% SDS, and 1.25 μL of Proteinase K, followed by phenol/chloroform extraction and ethanol precipitation.

The purified Tn5-cleaved products from both assays underwent size selection using 0.65×/1.2× SPRIselect (Beckman Coulter) to remove genomic DNA contamination. To prepare the sequencing library, half of the total products were amplified for 3 cycles in the first-round PCR (primers: P5-ctMPRA-amp02/03-i# and P7-ctMPRA-amp02/03-UMI (Supplementary Data 11; Supplementary Fig. 1c); cycling

program: 98 °C for 1 min; 3 cycles of 98 °C for 10 s, 68 °C for 30 s, and 72 °C for 1 min; final extension at 72 °C for 5 min) to attach P5 and P7 flow cell adapters and UMIs. The amplicons were then purified using 1.0× AMPure XP (Beckman Coulter). qPCR was performed using the purified product, and the cycle number at which the reaction reached 80% of the plateau was determined as the cycle number for the second-round PCR. The second-round PCR was performed for 17–24 cycles using the primer set P5/P7. To estimate the count of each CRE inserted into genomic DNA, we used 1 μg (the pilot library and the HepG2 library) or 4 μg (the WTC11 library) of gDNA obtained from the lentiMPRA AllPrep extraction and performed the same PCR amplification to construct an additional sequencing library. The sequencing libraries were pooled and sequenced on a iSeq (the pilot library) or NextSeq Mid output 150-cycle kit (the HepG2 and WTC11 library) flowcell with custom primers (Read 1: pLSmP-ass-seq02/03-R1, Read 2: pLSmP-ass-seq02/03-R2, i7 index: ctMPRA-seq02/03-UMI, i5 index: ctMPRA-seq02/03-idx; Supplementary Data 11; Supplementary Fig. 1d) using the following setting: 72 + 72 + 15 + 8 bp.

### Sequencing data processing pipeline

**lentiMPRA data processing.** The analysis of CRE-barcode associations and DNA/RNA barcode sequencing data was performed using MPRA-flow v2.3.5[12] with minor modifications. For CRE-barcode association data, FASTQ files were generated using bcl2fastq with the following parameters: "--no-lane-splitting --minimum-trimmed-read-length 0 --mask-short-adapter-reads 0 --use-bases-mask Y145n,Y15,I10,Y145n". These FASTQ files were used as input for "association.nf" in MPRAflow, aligning reads to the designed library sequences with the following parameters:"--min-cov 3 --mapq -1 --cigar 100 M". Since our libraries, particularly the WTC11 library, were highly sensitive to single-nucleotide mismatches, we modified the alignment strategy by replacing "bwa mem" with "bwa aln -n 0" to exclude all reads containing any mismatches. For DNA/RNA barcode quantification, FASTQ files were generated from barcode sequencing data using bcl2fastq with the following settings: "--use-bases-mask Y15n,Y16,I10,Y15n". These FASTQ files were then processed with count.nf in MPRAflow using the parameters: "--bc-length 15 --umi-length 16 --thresh 5". The resulting DNA/RNA barcode count files for each replicate were subsequently used in downstream analyses.

**ATAC and CUT&Tag data processing.** For ATAC and CUT&Tag sequencing data, FASTQ files were generated using bcl2fastq with the following parameters: "--no-lane-splitting --minimum-trimmed-read-length 0 --mask-short-adapter-reads 0 --use-bases-mask Y72,Y15,I8,Y72". This command generated paired-end reads of CREs as R1 and R3, and UMI reads as R2. Using these FASTQ files, we first associated the CRE with UMIs using fastp v0.23.2[32] with the following parameters: "-i $R1/$R3 -I $R2 --umi --umi_loc=read2 --umi_len=15 -w 1 -Q -A -L -G -u 100 -n $UMI_LENGTH -Y 100. This command appended the UMI to the end of the first part of the read header for R1/R3. Next, a consensus sequence was generated from the paired-end reads using fastq-join v1.3.1[33]. All consensus sequences were then aligned to the designed sequences using BWA[34] with the parameter: "bwa aln -n 0". The resulting BAM file was processed to remove PCR duplicates using UMI-tools v1.1.4[35] with the following parameters: "dedup --per-gene --per-contig --umi-separator = ":"". Finally, UMIs per CRE were counted from the deduplicated BAM file using samtools[36] idxstats (v1.11). This process was performed separately for each replicate.

**Replicates, normalization and activity scores.** Activity scores were calculated by dividing the enriched counts of CREs (or barcodes) from each assay by the counts detected from genomic DNA. For lentiMPRA data, the activity score for each CRE was calculated as the ratio of "rna_count" to "dna_count" of barcodes. For ATAC and CUT&Tag data, activity scores were determined by dividing the number of UMIs per

CRE enriched by the number of UMIs per CRE amplified from genomic DNA. To ensure comparability of activity scores across different assays, replicates, and libraries, we first performed counts per million (CPM) normalization. We then calculated a scaling factor f using the trimmed mean of M-values (TMM) normalization method[37], based on the enriched counts and genomic counts of random genomic sequences, which served as a negative control. We assumed that these sequences exhibited no variability between the enriched counts from assays and the genomic counts. All enriched counts for the library sequences were then divided by the calculated scaling factor f. This normalization process ensured that the activity scores were adjusted so that the expected $\log_2(Activity)$ of the negative control sequences was 0 across all samples. To combine the data from all three replicates, we followed the method described by Wang et al.[38], which first summarizes the enriched and genomic counts within each replicate and then calculates the activity score by dividing the enriched count by the genomic count. For regression analysis, we used the activity scores from each replicate.

### Endogenous activity of CREs in the pilot library

The endogenous activities of CREs from genomic sequences in the pilot library were estimated using whole-genome epigenetic assay data from ENCODE[13]. We followed our previous method (Agarwal et al.)[10,11] to obtain and compute the consensus signal from these data. In brief, we extracted three bigWig files for H3K27ac ChIP-seq data (ENCFF084DIM, ENCFF515WSE, ENCFF759SNY) and five bigWig files for ATAC-seq data (ENCFF622FRD, ENCFF024GLW, ENCFF240VVR, ENCFF782GKX, ENCFF029XKY) of HepG2 cells from the hg38 assembly in ENCODE. For each CRE, we first extended the original 100 bp genomic region to 500 bp to mitigate positional biases and then calculated the mean bigWig signal from the corresponding genomic region using bigWigAverageOverBed[39]. All data were log-transformed, and multiple replicates corresponding to the same CRE were averaged to compute the consensus signal for each assay. These signals were then compared with the $\log_2(Activity)$ of e2MPRA.

### Analysis of HepG2 synthetic enhancer library

All analyses were performed using in-house Python scripts based on statsmodels[40] v0.14.2 and scikit-learn[41] v1.2.2. The difference between the two template types was considered as an additional replicate, resulting in a total of $2 \times 3$ replicates for each sequence. Of the nine TF motifs used, sequences containing NR2F2 motif were either absent or present at very low counts. From our previous experience, we have repeatedly observed that sequences with long stretches of polyC or polyG bases are often not properly assigned to barcodes during the CRE–barcode association sequencing step. Although the exact cause is unknown, it is likely due to synthesis or PCR errors. In our data, CREs containing a single NR2F2 motif were successfully sequenced, whereas those with two or more NR2F2 motifs were not detected in the lentiMPRA dataset (Supplementary Data 6). Moreover, almost none of the NR2F2-containing sequences were detected in the genomic count, ATAC-seq, or H3K27ac CUT&Tag data (Supplementary Data 7); therefore, we excluded these cases from downstream regression analyses.

To analyze the homotypic amplification of each epigenetic activity score, we calculated Spearman's correlation and its P-value between $\log_2(Activity)$ and the count of motifs for each motif and assay. All P-values were adjusted using the Benjamini–Hochberg method[42] (FDR = 0.05).

To evaluate potential synergistic effects between two TFs, we performed pairwise multiple linear regression analysis for each TF pair. The activity score was modeled as a function of the TF motifs present in each sequence. The observed $\log_2(Activity)$ of Class 1 and Class 2 sequences was used as the response variable, while the individual TF motif count was used as the predictor variable. Additionally, we introduced an interaction term k, defined as: k = {0 | Class 1, 1 or 2 |

Class 2}. The regression model was expressed as follows:

$$\log_2(Activity) \sim count_{TF1} + count_{TF2} + k \tag{1}$$

Multiple linear regression was performed for all TF pairs, and the statistical significance of the coefficient of k was assessed by applying multiple comparison corrections to the P-values. Synergistic effects were considered significant if the adjusted P-values met the predefined significance threshold (FDR = 0.01).

For Class 3 sequences, we first tested whether changes in TF motif order significantly affected epigenetic activity using one-way analysis of variance (ANOVA). The analysis was performed across all 70 unique motif sets generated by selecting 4 TF motifs out of 8 ($_8C_4 = 70$), with each set comprising 24 permutations (4! = 24). For each motif set, one-way ANOVA was conducted using the activity scores of the 24 permutations as input. Statistical significance was determined from ANOVA P-values, followed by Benjamini–Hochberg correction.

To further investigate how the position of each motif relative to the minimal promoter (mP) influences activity, we conducted a positional enrichment analysis. Specifically, we selected the top 200 and bottom 200 sequences based on transcriptional activity from all permutations and calculated the frequency of each TFBS at each of the four positions (position 1 = farthest from mP, position 4 = closest to mP) as odds ratios. Enrichment was assessed using a hypergeometric test, and P-values were adjusted using the Benjamini–Hochberg method.

### Analysis of WTC11 enhancer perturbation library

**Single nucleotide substitution subset analysis.** To infer the effects of single nucleotide variants, we followed previous studies[43,44] with minor modifications. This approach enables formal testing of variant effects and provides p-values that quantify their statistical significance. We fitted a linear regression model of the form:

$$\log_2(enriched\ count) \sim \log_2(inserted\ count) + N + offset \tag{2}$$

where enriched count refers to the number of unique molecular identifiers (UMIs) detected in the RNA, ATAC and CUT&Tag fragments, inserted count refers to the number of UMIs detected in the gDNA, representing the initial insertion frequency, and N represents binary indicators for mutated nucleotides and their positions. The estimated coefficients of N and their P-value were reported as the effects for each variant.

**6 bp window perturbation subset analysis.** To identify significant functional sites from the 6 bp window perturbation result, we adapted the Canny edge detection method[45] to the one-dimensional data. This simplified Canny method was used to detect regions where perturbations induced abrupt changes in activity scores, identifying these regions as functional sites.

i) Positional effect calculation.

For each position $i$, we calculated the positional effect using the median activity score of six sequences perturbing that position (due to the 6 bp window perturbation design). To assess the statistical significance of the positional effect, we computed the median absolute deviation (MAD) score. The MAD score was calculated using the wild-type (WT) activity score as the mean absolute deviation reference. However, in the ATAC and H3K27ac assays, the number of UMIs for the WT was abnormally higher than that of the variants across all target CREs—around 10,000 UMIs per WT sequence, compared to only hundreds for the variants (Supplementary Data 9). Therefore, instead of using the WT activity score directly, we estimated its reference value by taking the median of single nucleotide substitution results.

ii) Edge detection and functional site identification.

The computed MAD scores were smoothed using a Gaussian filter, and their derivatives were obtained for edge detection. Based on these derivatives, we applied the non-maximum suppression (NMS) technique to enhance edge localization. For hysteresis thresholding, we classified the detected edges as follows:

1. Strong edges: Local maxima identified through NMS and exceeding the median of derivative scores
2. Weak edges: Values exceeding the median of derivative scores.
3. Weak edges connected to strong edges were recursively clustered and reclassified as strong edges.

Through this process, regions with substantial relative changes in activity scores were identified as strong edges.

iii) Peak detection and functional site estimation.

To further suppress noise, we selected MAD score peaks based on the extrema of smoothed values, where absolute values exceeded 0.75. Finally, functional site regions were defined as adjacent regions between peak candidates and strong edges.

## Statistics & reproducibility

Statistical analyses for e2MPRA results are detailed in each analysis section of the Methods. Statistical tests for the sequencing data are described in the corresponding figure legends. No statistical method was used to predetermine sample size. For lentiMPRA analyses, only barcodes observed in both DNA and RNA within a replicate were retained, and elements with fewer than 5 unique barcodes were excluded. For the HepG2 synthetic enhancer library, sequences containing the NR2F2 motif were excluded due to absent or very low genomic counts. For the WTC11 enhancer perturbation library, one active CRE (seq6846_R) was excluded due to insufficient read coverage, and downstream analyses were restricted to CREs/variants meeting the read-coverage and detection criteria described in the Results and Methods. The experiments were not randomized. Reproducibility was assessed by comparing replicate measurements (e.g., correlations between replicates). Code and primary sequencing datasets are publicly available as described in the Code availability and Data availability sections.

## Reporting summary

Further information on research design is available in the Nature Portfolio Reporting Summary linked to this article.

## Data availability

The e2MPRA sequencing data generated in this study, including association barcode sequencing data and barcode sequencing data for lentiMPRA, as well as ATAC-seq and CUT&Tag-seq data, have been deposited in the DDBJ database under accession code PRJDB39977 and at the Zenodo repository[46,47]. Publicly available H3K27ac ChIP-seq data (ENCFF084DIM, ENCFF515WSE, ENCFF759SNY) and ATAC-seq data (ENCFF622FRD, ENCFF024GLW, ENCFF240VVR, ENCFF782GKX, ENCFF029XKY) of HepG2 were downloaded from ENCODE portal. Source data are provided in this paper.

## Code availability

The code used for data processing and analysis in this study is available at: https://github.com/ziczhang/e2MPRA_analysis and at the Zenodo repository[48].

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

## Acknowledgements

This work was supported by the World Premier International Research Center Initiative (WPI), MEXT Japan, MEXT KAKENHI Grant Numbers JP24K02004 (F.I.), JP24K18101 (Z.Z.), and AMED under Grant Number JP24gm7010002 (F.I.). This work was funded in part by the National Human Genome Research Institute grant numbers 1R21HG010683 (N.A.), 1UM1HG009408 (N.A.) and 1UM1HG011966 (N.A.). We thank the Single-Cell Genome Information Analysis Core (SignAC) at WPI-ASHBi, Kyoto University, for their support. The WTC11 cell line was kindly provided by Dr. Bruce R. Conklin (The Gladstone Institutes and UCSF).

## Author contributions

F.I. and N.A. conceived the study. Z.Z., I.G and F.I. designed the e2MPRA library. Z.Z. and F.I. performed experiments. Z.Z. analyzed data. Z.Z., F.I. and N.A. wrote the paper. I.G. and G.B. assisted with manuscript writing and editing.

## Competing interests

F.I. receives funding from Relation Therapeutics. N.A. is a Cofounder and on the scientific advisory board of Regel Therapeutics Inc. N.A. received funding from BioMarin Pharmaceutical Inc. The remaining authors declare no competing interests.
