## [Transparent Peer Review file · Nature Communications]

Simultaneous epigenomic profiling and regulatory activity measurement using e2MPRA

Corresponding Author: Dr Fumitaka Inoue

Version 0:

Reviewer comments:

Reviewer #2

(Remarks to the Author)

Zhang et al present a new approach e2MPRA for simultaneous epigenomic profiling and measuring regulatory activity. E2MPRA builds on the authors previous work on lentivirus based MPRA by incorporating epigenetic profiling in addition to measuring RNA from the integrated sequence. This is an interesting method that could be used to identify factors that lead to changes in chromatin state without activating the regulatory element as exemplified in the present study as well. To show the utility of e2MPRA the authors present two use cases: 1) testing synthetic regulatory elements with binding motifs for nine transcription factors (individual or in combination) in HepG2 cells and 2) the impact of genetic variants on enhancer activity and epigenetics marks at the regulatory element.

While the presented approach provides potentially important insight into CRE function, there are several points that need to be addressed:

Including epigenetic information is major advance of the presented technologies. However, it seems not ideal that a) the correlation with regular epigenetic profiling and b) reproducibility between replicates (0.4-0.5) is low. How do the authors envision that this could be improved? Is the chromatin state of the elements affected by the integration site in the genome?

Inactive synthetic sequences have higher levels of active chromatin marks than active genomic sequences. What is the underlying reason? Could this impact the assay performance, e.g. the difference between active and inactive at genomic regions seems larger than between active and inactive synthetic regions (Fig. 2d)? Could the difference be increased? Is this part of the reason for the noisy signals when testing the impact of genetic variants (Fig 5c,d)?

The authors mention that NR2F2 motif needed to be excluded based on sequence based on high GC content. CTCF motif also has high GC content. Was it also impacted or not?

Related to Fig 3:

How to explain that ATAC and lentiMPRA for HNF1A is increased but unchanged/decreased levels of H3K27ac? Similar for XBP1 increase lentiMPRA leads to unaltered/lower H3K27ac?

How do the MPRA results compare to CREs in HEPG2 cells? Do endogeneous CRE show similar changes in epigenetic marks depending on presence of different TF motif and motif combination?

Activity of TFs combinations with REST is comparable to REST only. However, these interactions are stated as synergistic effects (red label). How is the function of e.g. REST and together with the activator CEBPA synergistic? Could a possible explanation be that REST establishes a chromatin state incompatible with activation by the other factors?

PPARA establish chromatin to facilitate expression. What is the function/mechanistic role of the other factors in the tested model (no 3D genome architecture, no enhancer-promoter-looping) to induce gene expression?

(Remarks on code availability)

Reviewer #3

(Remarks to the Author)

In this study, Zhang et al. developed a new method they called e2MPRA that utilizes lentivirus-based MPRA to enrich for the integration of specific CREs into the genome followed by Cut&Tag or ATAC-seq targeted specifically for these sequences. While previous methods focused on either measuring regulatory activity or epigenetic modification, this new method measures simultaneously regulatory activity, protein binding and epigenetic modification of thousands of candidate CREs.

1. They first tested the method on 400 100bp enhancers (including positive and negative controls). The results show that e2MPRA measures accurately the activity of the enhancer, with ATAC-seq and H3K27ac CUT&Tag activities across the library correlating with lentiMPRA activity (but correlation is only 0.64-0.67). In fact, there is a moderate correlation between epigenetic activity as measured by e2MPRA and endogenous ATAC-seq and H3K27ac. I think this is not only driven by difference between the epigenetic activity measured by e2MPRA and endogenous methods, but rather, by the fact that combinatorial signal of different epigenetic modifications could better explain enhancer activity than single epigenetic modifications alone (e.g., PMID:34749786) or that other epigenetic modifications beyond H3K27ac could predict enhancer activity (e.g., PMIDs:37308596,27089178). Authors could discuss this in the manuscript.

2. They use e2MPRA to measure binding of different TFs, enhancer activity and epigenetic profiles at these enhancers. They find that HNF1A and ONECUT1 can potentially open chromatin, which is very interesting and supports reports from the literature. However, this is not the case for FOXA1 (sometimes they refer to this as FOX1A, but I assume that is only a typo). This is interesting since previous studies have reported FOXA1 as a pioneer factor (PMID:27259199), but also some studies found that it might have only moderate capacity to bind closed chromatin and only at the strongest sites (e.g., PMID:37486787). These new results seem to support the latter model, and authors could discuss this in the manuscript.

3. PPARA seems to facilitate transcriptional activation of other TFs by mediating the enhancer epigenetic activity, while REST acts as a repressor and the closer it is to the promoter the stronger the repression is. The latter is known and is nice to see that recapitulated. What maybe the authors could explain more is how REST does not seem to achieve this by closing the chromatin or depleting H3K27ac. Maybe they discussed this, but I missed it.

4. They tested the effects of variants on activity, ATAC-seq and H3K27ac and found that disruptions of POU5F1::SOX2 and ETS motifs in significantly decreased ATAC-seq and H3K27ac activities, which is interesting and supporting of what is expected for POU5F1::SOX2 and ETS. One aspect that was not clear to me it is why the authors used a linear regression model (I am not saying it is wrong, just that it was not explained).

Overall, I think this is an important paper that should be published in Nature Communications, given the authors address my comments above.

(Remarks on code availability)

Version 1:

Reviewer comments:

Reviewer #2

(Remarks to the Author)

The authors have addressed all my comments in the revision. I recommend publication of the manuscript.

(Remarks on code availability)

Reviewer #3

(Remarks to the Author)

The authors addressed my comments.

(Remarks on code availability)

RESPONSE TO REVIEWERS

We thank the reviewers for the great comments, which have significantly improved our revised manuscript. Below, we provide a point-by-point response to these reviews.

Reviewer #2

Zhang et al present a new approach e2MPRA for simultaneous epigenomic profiling and measuring regulatory activity. E2MPRA builds on the authors previous work on lentivirus based MPRA by incorporating epigenetic profiling in addition to measuring RNA from the integrated sequence. This is an interesting method that could be used to identify factors that lead to changes in chromatin state without activating the regulatory element as exemplified in the present study as well. To show the utility of e2MPRA the authors present two use cases: 1) testing synthetic regulatory elements with binding motifs for nine transcription factors (individual or in combination) in HepG2 cells and 2) the impact of genetic variants on enhancer activity and epigenetics marks at the regulatory element.

We thank the reviewer for the valuable comments and for recognizing the importance of our work.

While the presented approach provides potentially important insight into CRE function, there are several points that need to be addressed:

Including epigenetic information is major advance of the presented technologies. However, it seems not ideal that a) the correlation with regular epigenetic profiling and b) reproducibility between replicates (0.4-0.5) is low. How do the authors envision that this could be improved? Is the chromatin state of the elements affected by the integration site in the genome?

We thank the reviewer for this comment. We agree that the concordance of e2MPRA with the endogenous signals and replicate correlations leave room for improvement. In the revised version, we have tried to address these comments. For comment a), the low correlation with regular epigenetic profiling (related to Fig. 2d) mainly stems from the difference of the methods for how tagged fragments are detected; i.e. e2MPRA uses specific primers to quantify epigenetic modifications by counting how many times each element has been tagged. In contrast, genome-wide Cut&Tag and ATAC-seq measure the enrichment based on the number of unique reads in each region. However, more importantly, the use of specific primers in e2MPRA allows us to perform a fair comparison between sequence variants in the library, even though it will not quantitatively capture the endogenous signals. To further clarify this, we now mention this issue in our revised manuscript:

“This could be due to a technical aspect with e2MPRA where specific primers are used to count how many times each element has been tagged, while genome-wide Cut&Tag and ATAC-seq measure the enrichment based on the number of unique reads in each region.”

In addition, we also added the following text to the discussion:

“We further observed a low correlation between e2MPRA and endogenous epigenetic signals. This could be due to differences in measurement of the two methods, i.e. e2MPRA activities were detected by specific primers and measured in the vector context, while endogenous activities were detected by random fragmentation. The use of specific primers in e2MPRA allows for unbiased comparisons between sequence variants within the library. Therefore, e2MPRA could be more suitable for the parallel quantitative comparison of enhancer activities among sequence variants rather than for recapitulating endogenous epigenetic modifications.”

In terms of the reviewer’s comment b), the low reproducibility between replicates; this could be improved by using easier-to-infect cells (see also WTC11 (Fig. S6d) that is harder to infect and showed lower correlation than HepG2 (Figs. S2f and S3d), which is easier to infect, and adding more sequencing reads and more experimental replicates. To address this, we aggregated barcode counts across replicates as explained in the article:

“These moderate correlations are consistent with the way that the present method directly counts CRE fragments using UMIs, leading to greater error propagation when calculating activity, whereas lentiMPRA uses the average of multiple barcodes. To mitigate noise, we summed the counts across replicates and then computed activity by dividing the total enriched count by the total inserted count for each element.”

In this way, we were able to detect known and unknown epigenetic functions of transcription factors (Figs. 3b and 5f); therefore, we are confident that our experimental approach and data analysis methods are robust enough.

Inactive synthetic sequences have higher levels of active chromatin marks than active genomic sequences. What is the underlying reason? Could this impact the assay performance, e.g. the difference between active and inactive at genomic regions seems larger than between active and inactive synthetic regions (Fig. 2d)? Could the difference be increased? Is this part of the reason for the noisy signals when testing the impact of genetic variants (Fig 5c,d)?

We apologize for the confusion regarding the category labeling. Both the “active” and “inactive” synthetic sequences were taken from our previous study (Smith et al. *Nature Genetics* 2013) and composed of multiple binding sites of transcription activators. The “active” and “inactive” sets correspond to the top 50 and the bottom 50 sequences based on the MPRA activity, respectively. Hence, although labeled “inactive”, those sequences still contain TF binding sites and potentially carry epigenetic activities. The figure below reflects differences in the compositions of active binding sites between these two subsets. In it, we show for example that the “inactive” sequences include a similar number of PPARA binding sites as “active” sequences. This TF is known to be involved in histone modification and pioneering activities and could thus lead to open chromatin also in the “inactive” sequences. In contrast, the active sequences contain more TFBS copies of HNF1A for example, which is a strong activator in HepG2 cells (Smith et al. *Nature Genetics* 2013).

To avoid confusion, we re-labeled these groups as “top 50” and “bottom 50” throughout the manuscript. We should also note that this library is independent from other libraries and does not cause noisy signals or impact on other results.

We added the following text in the revised article and also the above figure as Extended Data Fig 2g:

“As expected, the bottom 50 synthetic sequences showed lower lentiMPRA activity than active genomic sequences; however, they displayed relatively high ATAC-seq and H3K27ac signals. This observation is consistent with the fact that all synthetic sequences were composed of active TF motifs, and that the bottom 50 sequences are enriched for binding sites of pioneer factors (e.g., ONECUT1, PPARA), rather than transcriptional activators (e.g., HNF1A, XBP1) (Extended Data Fig. 2g).”

The authors mention that NR2F2 motif needed to be excluded based on sequence based on high GC content. CTCF motif also has high GC content. Was it also impacted or not?

In this study, we unexpectedly failed to retrieve sequences containing NR2F2 motif (CCCCCTGACCTTTGCCCTGCC) in the library, whereas successfully obtained CTCF (CGGCCACCAGGGGCGCCA) sequences. In our previous experiences, we have repeatedly observed that sequences with many polyC or polyG bases were not properly assigned to barcodes during the CRE–barcode association sequencing step. Although we are not certain of the exact cause, it is likely due to synthetic or PCR errors. Barcode counts per CRE are provided in Supplementary Table S6 (column n_obs_bc). In our data, CREs containing a single NR2F2 motif were successfully sequenced, but CREs with two or more NR2F2 motifs were not; therefore, we excluded these cases from downstream regression analyses. CTCF did not have this issue and was included in the analysis. To address this, we have added the following text in the Method section:

*“Of the nine TF motifs used, sequences containing the NR2F2 motif were either absent or present at very low counts. From our previous experience, we have repeatedly observed that sequences with long stretches of polyC or polyG bases were not properly assigned to barcodes during the CRE–barcode association sequencing step. Although we are not certain of the exact cause, it is likely due to synthesis or PCR errors. In our data, CREs containing a single NR2F2 motif were successfully sequenced, whereas those with two or more NR2F2 motifs were not detected in the lentiMPRA dataset (**Supplementary Table 6**). Moreover, almost none of NR2F2 containing sequences were detected in the genomic count, ATAC-seq and H3K27ac CUT&Tag (**Supplementary Table 7**); therefore, we excluded these cases from downstream regression analyses.”*

Related to Fig 3:

How to explain that ATAC and lentiMPRA for HNF1A is increased but unchanged/ decreased levels of H3K27ac? Similar for XBP1 increase lentiMPRA leads to unaltered/lower H3K27ac?

We thank the reviewer for this great comment. Indeed, H3K27ac signals remained unchanged regardless of the number of CEBPA, FOXA1, HNF1A and XBP1 motifs (Fig. 3b, Extended Data Fig. 4). Possible reasons that explain this observation could be: 1) Improving e2MPRA method (adding more reads and more replicates) are required to detect H3K27ac signal. 2) these factors activate gene transcription by directly interacting with gene promoters without being involved in H3K27ac modification.

To address this, we have added the following text to the discussion:

*“While CEBPA, FOXA1, and XBP1 increase lentiMPRA activity along with the number of their motifs, we did not observe epigenetic changes for them (**Fig. 3b, Extended Data Fig. 4**). This could be due to these factors being responsible for activating gene transcription by direct interaction with gene promoters without epigenetic modification and/or not having a role related to H3K27ac. Another reason could be due to the limitation of e2MPRA technology, requiring additional reads and replicates to obtain more robust epigenetic results. Thus, we cannot exclude the possibility that these factors are involved in epigenetic modifications.”*

How do the MPRA results compare to CREs in HEPG2 cells? Do endogenous CRE show similar changes in epigenetic marks depending on presence of different TF motif and motif combination?

We thank the reviewer for this suggestion. We have investigated endogenous CREs and found that the majority of CREs consist of combinations of multiple TF motifs, rather than containing a single isolated TF motif. This makes it difficult to quantify the function of individual factors and their cooperative effects in an endogenous system.

The figure below shows an analysis similar to Fig. 3c but performed on endogenous CREs. We reanalyzed the lentiMPRA data from Vikram et al., *Nature* 2025, which measured transcriptional activities of endogenous CREs in HepG2 cells, and selected sequences that contained at least one of the TFBSs examined in our study. For example, in our e2MPRA, PPARA exhibited strong synergistic effects when

combined with other transcription factors (a), whereas in the endogenous data, combinations with other factors showed almost no change in activity distribution compared to CREs containing a single TFBS (b). Similarly, although REST functioned as a strong repressor in our e2MPRA, this effect was not observed in the endogenous data. We think that these observations do not indicate inconsistency between the two systems; rather, they reflect the inherent complexity of endogenous CREs, which contain multiple TFBSs and whose activities are highly context dependent. This complexity makes it difficult to isolate and quantify the contribution of each single motif.

Figure Legend: Synergistic transcriptional effects when combining PPARA or REST motifs with other TF motifs for synthetic constructs (a) and endogenous CREs (b). Transcriptional activities of homotypic arrangements (two identical motifs) were compared with heterotypic arrangements (one motif combined in a 1:1 ratio). For synthetic constructs, statistically significant synergistic effects ($FDR < 0.01$) are indicated by a red background, and non-significant combinations are shown in blue, as in the main Figure 3c. In Figure 3c, we compared homotypic arrangements of four identical motifs with heterotypic arrangements of two motifs combined in a 2:2 ratio. However, because endogenous CREs containing four identical motifs could not be identified, we compared homotypic arrangements of two identical motifs with heterotypic arrangements combining one motif in a 1:1 ratio. The corresponding synthetic constructs were analyzed in the same manner for consistency.

By contrast, our synthetic framework places TFBSs on an inactive template sequence, thereby removing genomic contextual effects and enables precise measurement of each factor's contribution and their combinatorial interactions. We therefore view our results as directly informative for understanding the complex cooperativity among TFBSs that operates within endogenous CREs.

Together, these results highlight the importance of analyzing synthetic enhancers composed of TF motifs to understand molecular epigenetic function of each TFs using e2MPRA, rather than just investigating genomic enhancers.

Activity of TFs combinations with REST is comparable to REST only. However, these interactions are stated as synergistic effects (red label). How is the function of e.g. REST and together with the activator CEBPA synergistic? Could a possible explanation be that REST establishes a chromatin state incompatible with activation by the other factors?

We thank the reviewer for catching this mistake. The term "synergistic" should not be used for REST's repressive effect. We rephrased a sentence in the Fig.3c legend and the Results section as below:

"c. Representative examples showing synergistic transcriptional effects when combining PPARA motif with other TF motifs, as well as the repressive effect of REST over other motifs (Class 2)."

"In addition to the synergistic effects, we also observed repressive effects mediated by REST. REST significantly reduced lentiMPRA activity when combined with any other TF except ONECUT1 (Figs. 3c-d). This suggests that REST represses transcription in the presence of adjacent activators."

In this study, we found that REST functions as a strong repressor even when present alongside a transcriptional activator (Fig. 3c). We also added the following text in the discussion (see also the third comment from the reviewer 3):

"REST is a well-characterized transcriptional repressor that mediates silencing through recruitment of chromatin modifiers, including HDACs, LSD1, G9a/GLP, and PRC1/2. In our assay, REST repressed transcription when present together with transcriptional activators (Fig. 3c), suggesting that it exerted the strongest effect among the factors tested alongside it. By contrast, we did not detect a significant depletion of H3K27ac and closed chromatin for REST (Extended Data Fig. 4). This could be due to a variety of factors, including noisy measurements in this specific e2MPRA. Further technical improvement may be required to better understand its molecular function."

PPARA establish chromatin to facilitate expression. What is the function/mechanistic role of the other factors in the tested model (no 3D genome architecture, no enhancer-promoter-looping) to induce gene

expression?

We thank the reviewer for this comment. In the previous version of the manuscript, we discussed the function of other factors in the discussion:

“We showed that CEBPA, FOXA1, HNF1A, and XBP1 motifs increased transcriptional activity (Extended Data Fig. 4), consistent with previous results [11,14] and their known roles as transcriptional activators. In addition, HNF1A and ONECUT1, previously characterized as pioneer factors [16], were shown to increase chromatin accessibility”

We have also added the below text in the discussion, according to the reviewer’s suggestion.

*“While CEBPA, FOXA1, and XBP1 increase lentiMPRA activity along with the number of their motifs, we did not observe epigenetic changes for them (**Fig. 3b, Extended Data Fig. 4**). This could be due to these factors being responsible for activating gene transcription by direct interaction with gene promoters without epigenetic modification and/or not having a role related to H3K27ac. Another reason could be due to the limitation of e2MPRA technology, requiring additional reads and replicates to obtain more robust epigenetic results. Thus, we cannot exclude the possibility that these factors are involved in epigenetic modifications.”*

Reviewer #3

In this study, Zhang et al. developed a new method they called e2MPRA that utilizes lentivirus-based MPRA to enrich for the integration of specific CREs into the genome followed by Cut&Tag or ATAC-seq targeted specifically for these sequences. While previous methods focused on either measuring regulatory activity or epigenetic modification, this new method measures simultaneously regulatory activity, protein binding and epigenetic modification of thousands of candidate CREs.

1. They first tested the method on 400 100bp enhancers (including positive and negative controls). The results show that e2MPRA measures accurately the activity of the enhancer, with ATAC-seq and H3K27ac CUT&Tag activities across the library correlating with lentiMPRA activity (but correlation is only 0.64-0.67). In fact, there is a moderate correlation between epigenetic activity as measured by e2MPRA and endogenous ATAC-seq and H3K27ac. I think this is not only driven by difference between the epigenetic activity measured by e2MPRA and endogenous methods, but rather, by the fact that combinatorial signal of different epigenetic modifications could better explain enhancer activity than single epigenetic modifications alone (e.g., PMID:34749786) or that other epigenetic modifications beyond H3K27ac could predict enhancer activity (e.g., PMIDs:37308596,27089178). Authors could discuss this in the manuscript.

We thank the reviewer for this suggestion. In our previous study (Agarwal et al. *Nature* 2025), we built a regression model that integrated multiple biochemical features at CREs (TF ChIP-seq, histone ChIP-seq, DNase-seq, and ATAC-seq) to predict lentiMPRA activity; the Pearson correlation was around 0.7-0.75 (Figure 3b in Agarwal et al. *Nature* 2025). As the reviewer suggests, this supports the idea that composite epigenomic features explain activity better than any single mark, while also indicating that epigenomic features alone have limited explanatory power. This underscores the value of our present approach, which quantifies each epigenomic feature separately. To address this, we added the following text to the revised discussion:

“It is generally thought that the combinatorial signal of multiple epigenetic modifications contributes to enhancer activity more than single epigenetic modifications alone. Indeed, our previous work integrating multiple biochemical features at CREs (TF ChIP-seq, histone ChIP-seq, DNase-seq, and ATAC-seq) enabled to predict lentiMPRA activity to a certain extent; the Pearson correlation was around 0.7-0.75 (Figure 3b in Agarwal et al. Nature 2025), indicating that epigenomic features alone have limited explanatory power. These findings further emphasize the value of our approach to analyze multiple epigenetic modifications separately.”

2. They use e2MPRA to measure binding of different TFs, enhancer activity and epigenetic profiles at these enhancers. They find that HNF1A and ONECUT1 can potentially open chromatin, which is very interesting and supports reports from the literature. However, this is not the case for FOXA1 (sometimes they refer to this as FOX1A, but I assume that is only a typo). This is interesting since previous studies have reported FOXA1 as a pioneer factor (PMID:27259199), but also some studies found that it might have only moderate capacity to bind closed chromatin and only at the strongest sites (e.g., PMID:37486787). These new results

seem to support the latter model, and authors could discuss this in the manuscript.

We thank the reviewer for this great suggestion! Consistent with this suggestion, the role of FOXA1 as a pioneer factor remains under active debate. In our assay, the number of FOXA1 sites was positively correlated with lentiMPRA activity, whereas neither ATAC-seq nor H3K27ac showed a positive correlation. These observations suggest that, under our conditions, FOXA1 behaved primarily as a transcription factor rather than as a pioneering factor. That said, its low significance (adjusted $p = 0.115$; Extended Data Fig. 4) could be due to assay noise; therefore, our data still cannot exclude its pioneer activity. We have added the following text in the discussion:

“The pioneering activity of FOXA1 is still being characterized. While early studies proposed that FOXA1 acts as a pioneer factor capable of binding to closed chromatin and facilitating subsequent transcriptional activation [23], more recent reports have suggested that its pioneering capacity may be limited to a subset of strong binding sites or may depend on cellular context [24]. In our e2MPRA assay, FOXA1 increased transcriptional activity without a corresponding increase in chromatin accessibility or H3K27ac modification (adjusted $p = 0.115$ in ATAC-seq; Extended Data Fig. 4), suggesting that under our experimental conditions, FOXA1 primarily functioned as a transcriptional activator rather than as a pioneer factor. Nevertheless, given the moderate statistical significance of the observed effects, we cannot rule out a pioneering activity for FOXA1.”

We also thank the reviewer for catching FOXA1 typos. All typos in the text are now fixed.

3. PPARA seems to facilitate transcriptional activation of other TFs by mediating the enhancer epigenetic activity, while REST acts as a repressor and the closer it is to the promoter the stronger the repression is. The latter is known and is nice to see that recapitulated. What maybe the authors could explain more is how REST does not seem to achieve this by closing the chromatin or depleting H3K27ac. Maybe they discussed this, but I missed it.

As the reviewer mentioned, REST represses transcription by recruiting chromatin modifiers, including HDACs, LSD1, G9a/GLP, and PRC1/2, which can lead to depletion of H3K27ac [25, 27]. In our assay, REST repressed transcription when present together with other transcription factors on a CRE, suggesting that it exerted the strongest effect among the factors tested. By contrast, we did not detect a significant depletion of H3K27ac, likely because the H3K27ac measurements were particularly noisy in this dataset. We have added the following text to the discussion:

*“REST is a well-characterized transcriptional repressor that mediates silencing through recruitment of chromatin modifiers, including HDACs, LSD1, G9a/GLP, and PRC1/2 [25, 27]. In our assay, REST repressed transcription when present together with transcriptional activators (**Fig. 3c**), suggesting that it exerted the strongest effect among the factors tested alongside it. By contrast, we did not detect a significant depletion of H3K27ac and closed chromatin for REST (**Extended Data Fig. 4**). This could be due to a variety of factors, including noisy measurements in this specific e2MPRA. Further technical*

improvement may be required to better understand its molecular function.”

4. They tested the effects of variants on activity, ATAC-seq and H3K27ac and found that disruptions of POU5F1::SOX2 and ETS motifs in significantly decreased ATAC-seq and H3K27ac activities, which is interesting and supporting of what is expected for POU5F1::SOX2 and ETS. One aspect that was not clear to me it is why the authors used a linear regression model (I am not saying it is wrong, just that it was not explained).

Our primary reason for using linear regression is continuity with our prior work, in which we estimated variant effects from saturation mutagenesis using linear models (Kreimer et al. 2022, *Nat. Comm.*, Ashuach et al. 2019, *Genome Biol.*). Linear regression also permits formal testing of the null hypothesis of no variant effect and provides p-values that quantify statistical support. This makes it straightforward to report, for each variant, the strength of evidence for its effect. To address this, we revised the following text in the Method section:

“To infer the effects of single nucleotide variants, we followed previous studies [45, 46] with minor modifications. This approach enables formal testing of variant effects and provides p-values that quantify their statistical significance.”

Overall, I think this is a important paper that should be published in Nature Communications, given the authors address my comments above.

We thank the reviewer for the supportive comments.